# A Survey on Low-Thrust Trajectory Optimization Approaches

**David Morante \***  , **Manuel Sanjurjo Rivo**  and **Manuel Soler**

Department of Bioengineering and Aerospace Engineering, Universidad Carlos III de Madrid,
28911 Leganés, Spain; msanjurj@ing.uc3m.es (M.S.R.); masolera@ing.uc3m.es (M.S.)
\* Correspondence: morantegonzalezdavid@gmail.com

**Abstract:** In this paper, we provide a survey on available numerical approaches for solving low-thrust trajectory optimization problems. First, a general mathematical framework based on hybrid optimal control will be presented. This formulation and their elements, namely objective function, continuous and discrete state and controls, and discrete and continuous dynamics, will serve as a basis for discussion throughout the whole manuscript. Thereafter, solution approaches for classical continuous optimal control problems will be briefly introduced and their application to low-thrust trajectory optimization will be discussed. A special emphasis will be placed on the extension of the classical techniques to solve hybrid optimal control problems. Finally, an extensive review of traditional and state-of-the art methodologies and tools will be presented. They will be categorized regarding their solution approach, the objective function, the state variables, the dynamical model, and their application to planetocentric or interplanetary transfers.

**Keywords:** low-thrust; hybrid optimal control; survey





## 1. Introduction

The exploration and exploitation of outer space play an essential role in the efficient functioning of modern societies. It contributes to advance scientific knowledge and technology innovation, to meet global challenges on Earth, as well as to generate substantial commercial revenues. Historically, space activities have been dominated by space-faring countries with large economies, a few big commercial enterprises, and little competition. However, over the past decade, the number of private and public players involved in space activities has increased. As a consequence, the space sector is undergoing fundamental transformations towards a more global and diverse ecosystem with a mix of government and commercial initiatives, a variety of contractors, and stiff competition. Meanwhile, missions of growing levels of sophistication, complexity, and scientific return are being proposed for the forthcoming years. Indeed, envisioned projects include megaconstellations of small satellites orbiting Earth, probes landing on the moons of outer planets, and human settlements being established on Mars.

In such a scenario, reducing the cost and schedule of accessing and using space without compromising quality and safety becomes a major goal. The potential benefits translate not only into economic gains for commercial space actors, yet into enhancing or enabling future scientific missions that cannot currently be accomplished due to budget or technological limitations. For such purpose, novel mission architectures and breakthrough technologies have become primary tools. Among them, the development of new commercial launch systems, the thriving generation of small satellites prompted by miniaturized but fully functional electronics, the recent advances in material sciences, and the implementation of distributed mission concepts will be shaping the global space sector during the next decades. On top of that, ambitious future projects will continue to benefit from the high fuel efficiency inherent to the well-stablished electric propulsion systems. Similarly, the use of gravity assisted maneuvers will remain as the chief means to lower the cost of reaching distant targets in the Solar System.

Notably, space mission analysis and design activities are also experiencing a paradigm shift to more rapid and cost-effective processes based on concurrent engineering principles. Contrary to traditional methods, in concurrent engineering the transfer trajectory and the mission architecture, i.e., mission planing, along with the spacecraft subsystems are designed simultaneously. Concurrent engineering approach is increasingly being used for the preliminary design of space missions. During this early period, scientists and decision-makers are interested in high-level trade-off analysis, i.e., exploring as many options as possible and assessing them against multiple, and often conflicting criteria. They are typically conducted on a short duration schedule with limited resources and input information. However, the success of this early phase has been demonstrated to drastically reduce resultant system life-cycle cost (up to 80%) and to increase the chances of a successful final design [1]. Moreover, at the Concurrent Design Facility from ESA it is claimed that the duration of the preliminary phase has been shortened from months to weeks by applying concurrent engineering practices. Therefore, multidisciplinary and automated software tools able to provide real-time performance trade-offs between the available options are highly desirable nowadays.

However, these requirements are difficult to be achieved in missions where the spacecraft has to travel from the injection orbit into its final destination using multiple gravity assists and/or electric propulsion. Mission designers have to optimize the transfer trajectory, the steering law of the electric engine, and/or the sequence of swing-bys that best accomplish the mission goals, while satisfying subsystems' constraints and operational restrictions. The selected path dictates the propellant expenditures and the time at which the spacecraft will be operational, thus utterly impacting mission feasibility, cost and return. Consequently, the optimization of low-thrust trajectories becomes an expensive process in terms of human and computer hours, where any automation, reduction in execution times, or increased flexibility and robustness are highly desirable to enhance the capabilities to design more ambitious and cost-effective missions. As a rule, it can be stated that better tools lead to better mission.

The optimization of trajectories involving chemical propulsion (CP) is a well-known problem and has been profusely studied in the literature; [2–6] provide a partial, but representative list of such prior works. Conversely, the optimization of trajectories involving low-thrust maneuvers are significantly more challenging. Note that the expression "low-thrust" encompasses a broad variety of quite different propulsion concepts, from electric propulsion (EP) to solar sail and tether techniques. In this article, low-thrust propulsion refers to EP only, unless noted otherwise. During the optimization of CP trajectories only a finite and small number of variables have to be considered, namely the number, magnitude and direction of the impulses. Meanwhile, low-thrust optimization requires the determination of a continuous steering law throughout the entire transfer, while satisfying subsystems' constraints and operational restrictions. The highly nonlinear and nonconvex dynamics, the space environment perturbations, and the existence of many local minima further complicates the optimization process [7]. Mission designers may be interested in determining the optimal number and sequence of gravity assisted maneuvers, or into including mission design decision-making and satellite subsystem design, as required by the concurrent engineering principles, as part of the solution. Therefore, searches over wide design spaces and solutions to complex combinatorial problems are demanded.

Classically, the optimization of low-thrust trajectories have been mathematically formulated as an Optimal Control Problem (OCP). This framework is limited to cases with continuous spacecraft dynamics, and with real variables and parameters. However, EP systems have two distinct discrete working modes (i.e., thrusting and coasting), and the dynamics, and consequently the trajectory, can be modeled as a hybrid dynamical system, i.e., a system with interacting continuous and discrete dynamics. The continuous dynamics determines the trajectory during the thrusting and coasting phases of the electric engine. Each phase represents a different working condition and consequently a different continuous dynamical description of the system. The discrete dynamics characterizes the

discontinuous behavior of the system such as the on/off switchings of the low-thrust engine or the effect of performing a gravity assisted maneuver. Additionally, mission planing and decision-making, which play a major role in concurrent engineering, are typically modeled as discrete or integer variables. In such scenario, the problem can be tackled as a Hybrid Optimal Control Problem (HOCP). General frameworks for the description of HOCPs and its corresponding mathematical formalism have been presented, e.g., by Branicky et al. [8] and Buss et al. [9]. Particular frameworks for space mission planning have been proposed by Chilan and Conway [10] and Ross and D' Souza [11].

In the literature, numerous numerical and analytical approaches have been reported to solve low-thrust trajectory optimization problems, based on either classical OCP or HOCP. One of the first attempts to categorize the available techniques was published in 1998 by Betts [12]. The author focused on OCP numerical techniques, namely direct and indirect methods, with boundary and path constraints and provided examples for general aerospace applications. Extending the work done by Betts, in 2005 Ross and D' Souza [11] included the newly developed approaches based on HOCP for mission planning. Later, in 2009 Rao [13] described typical methods and software tools that were developed for optimal trajectory generation. In 2012 Conway [14] described the advantages and disadvantages of the existing methods, and made an attempt to answer the question of what is the best extant numerical solution approach. Recently, Shirazi et al. [15] presented in 2018 an excellent review of models, objectives, approaches and solutions for spacecraft trajectory optimization, including both chemical and low-thrust propulsion system. They classified each of this elements and discussed their characteristics for solving these problems. Additionally, they provided a discussion on how to choose the best combination of models, objectives, and approaches for a given problem. However, they neglected the hybrid nature of the low-thrust trajectory optimization problem, the dynamic programming solution techniques, and the impact of the concurrent engineering principles on the newly available approaches.

The main purpose of this survey paper is to update, supplement and complete previous reviews on low-thrust trajectory optimization techniques. This summary also attempts to serve as a self-contained reference to the topic that includes state-of-the-art and classical methodologies for all those who are starting their research in low-thrust trajectory optimization. The goal is not only to describe and classify the available techniques, yet to identify the current research gaps and to propose possible approaches to tackle this gaps. In this article, we provide a general mathematical framework based on hybrid optimal control that is key to review the existing approaches and to develop new techniques customized for the concurrent engineering design of space missions.

The article starts by introducing the hybrid optimal control problem. Their elements, namely continuous and discrete state, controls, dynamics, and objective functions are analyzed in detail. Since they represent the key parts of every low-thrust trajectory optimization tool, they serve as the basis for discussion throughout the whole manuscript. Thereafter, the classical taxonomy of the numerical solution approaches, i.e., direct, indirect and dynamics programming methods, are briefly described. The approaches for continuous optimal control problems are presented first and extended later for their use in hybrid problems. Finally, an extensive review of traditional and state-of-the art methodologies and tools is presented. The analysis focuses, not only on methodologies proposed by the academia, yet also on tools developed by the industry. A set of look up tables incorporating a total of 90 references are provided to help the reader when searching for a specific tool along with a taxonomy: name of the tool, approach (direct, indirect or dynamics programming), solution (heuristic, gradient-based or hybrid), objective (single-objective or multiobjective), dynamics (perturbed restricted two-body problem or N-body problem), state representation (cartesian state, modified equinoctial elements or classical orbital elements) and the application (planetocentric, interplanetary or general). Since the subject is a vast one with a large literature, the research herein presented will be unapologetically incomplete.

## 2. Concurrent Engineering Requirements

Conventionally, elements of the space mission architecture are designed consecutively. However, this approach is being complemented and progressively replaced by concurrent engineering practices, especially during the preliminary design phase [16]. It involves the multidisciplinary design of the components collectively and in parallel. It pursues the goal of increasing competitiveness by decreasing lead-time while improving quality and cost. Nowadays, it is key to the low-cost design of space missions. Therefore, Team-X, formally called the Advanced Products Development Team, was created by the JPL (Jet Propulsion Laboratory, CA, USA) in 1995. It was followed by the Integrated Design Center (IDC) at Goddard Space Flight Center and COMPASS at Glenn Research Center. Similarly, the Concurrent Design Facility (CFD) from ESA (European Space Agency), was created in 1999 to rapidly perform feasibility studies for future missions. This concept has also been stablished in 2008 at the German Aerospace Center (DLR) Concurrent Engineering Facility (CEF), at the Satellite Design Office (SDO) of Airbus, and at the PASO office of CNES (National Centre for Space Studies, FR). Based on the concurrent engineering principles, the tools developed for low-thrust trajectory optimization must comply with the following set of requirements:

- Flexibility: high versatility to cope with a wide range of scenarios is demanded, as well as the ability to optimize discrete decision-making and mission planning.
- Robustness: the sensitivity to the input parameters has as low as possible.
- Speed: they have to be fast, since it is not possible to spend long computation times during concurrent design studies.
- Accuracy: they must provide meaningful results, yet high-fidelity is not required. An accurate trajectory will be required during the detailed design, once a mission candidate is selected.
- Automation: minimal user-interaction is desired to reduce man-power cost.
- Optimality: near-optimal solutions are deemed acceptable.

Consequently, thorough this paper, all the presented methodologies and approaches will be critically analyzed based on these criteria.

## 3. Multiobjective Hybrid Optimal Control

Mathematical frameworks for low-thrust trajectory optimization can be classified as continuous optimal control problems (COCP) or as hybrid optimal control problems (HOCP). Since HOCP are a generalization of COCP including discrete states, dynamics, and decision-making, they offer a much more flexible formulation, being the most suitable for concurrent engineering approaches. In this section, a formulation of the HOCP based on the one proposed by Buss et al. [9] is presented along with novel examples of their application to low-thrust trajectory optimization.

### 3.1. Hybrid Dynamical System

The state of a hybrid dynamical system is determined by the continuous state vector $\boldsymbol{x}(t) \in \mathcal{X} \subset \mathbb{R}^{n_x}$, which is constrained to be in the set $\mathcal{X}$ of permissible continuous states and the discrete state vector $\boldsymbol{q}(t) \in \mathcal{Q} \subset \mathbb{Z}^{n_q}$, which is constrained to be in the set $\mathcal{Q}$ of permissible discrete states. The system can be controlled by a continuous input vector $\boldsymbol{u}(t) \in \mathcal{U} \subset \mathbb{R}^{n_u}$, which belongs to the set $\mathcal{U}$ of permissible continuous controls, and by a discrete input vector $\boldsymbol{v}(t) \in \mathcal{V} \subset \mathbb{Z}^{n_v}$, which belongs to the set $\mathcal{V}$ of permissible discrete controls. Both input vectors (Input vectors can be also termed as control vectors, control inputs, control variables, controls or decision variables) can be dynamical variables or static parameters depending on whether they are time-varying or time-independent, respectively. Therefore, the evolution of the state vector with respect to the independent time variable $t \in \mathbb{R}$ is given by its hybrid dynamics as follows:

$$\dot{x} = f(x, q, u, v, t) \qquad \text{if} \quad s_j(x, q, u, v, t) \neq 0, \qquad j = 1, \ldots, n_s \qquad (1)$$

$$[x(t_i^+), q(t_i^+)] = \phi_j(x, q, u, v, t_i^-) \qquad \text{if} \quad s_j(x, q, u, v, t_i^-) = 0, \qquad j \in \{1, \ldots, n_s\}. \qquad (2)$$

The continuous behavior of the hybrid dynamical system is described by the set of differentiable equation $f : \mathcal{X} \times \mathcal{Q} \times \mathcal{U} \times \mathcal{V} \times \mathbb{R} \longrightarrow \mathbb{R}^{n_x}$, whereas the discontinuous behavior is characterized by the set of discrete event functions, which includes the $n_s$ discontinuity surfaces $s_j : \mathcal{X} \times \mathcal{Q} \times \mathcal{U} \times \mathcal{V} \times \mathbb{R} \longrightarrow \mathbb{R}$ and transition map functions $\phi_j : \mathcal{X} \times \mathcal{Q} \times \mathcal{U} \times \mathcal{V} \times \mathbb{R} \longrightarrow \mathcal{X} \times \mathcal{Q}$ for $j = 1, \ldots, n_s$. Discontinuity surfaces pose the condition that both state and controls must satisfy for a discrete event to be triggered. In case the discontinuous surface depends only on the state vector, it represents an autonomous event, whereas if it depends uniquely on the controls, it defines a controlled event.The times $t_i$ at which these events occur, are called event transition times. The successor states $x(t_i^+)$ and $q(t_i^+)$ just after a discrete event is given by the transition map functions. In case only the discrete state is changed after a discrete event, it is called a switching event, whereas if it is the continuous state experience a discrete jump, it is known as impulsive event. As long as all discontinuity surfaces $s_j(x, q, u, v, t) \neq 0$ for $j = 1, \ldots, n_s$, the system trajectory evolves continuously according to Equation (1).

Therefore, in a hybrid dynamical system, four basic types of discrete events can be found: autonomous switching, controlled switching, autonomous impulses, and controlled impulses [8]. Note that a general discrete event, as expressed in Equation (2), would comprise a combination of all of them. As an example, let us consider a hybrid system defined by a continuous state $x$, a discrete state $q$, and a discrete control $v$, and subject to Equations (1) and (2). Each type of discrete events have a different effect in the hybrid dynamics as it is illustrated in Figures 1 and 2. Further discussion is provided hereafter:

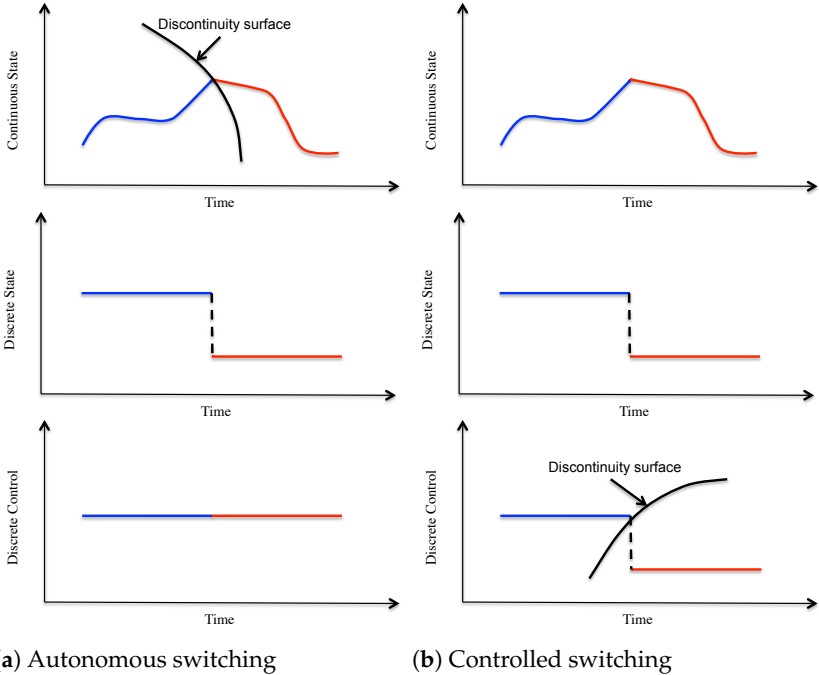

(**a**) Autonomous switching      (**b**) Controlled switching

**Figure 1.** Illustration of switching discrete events.

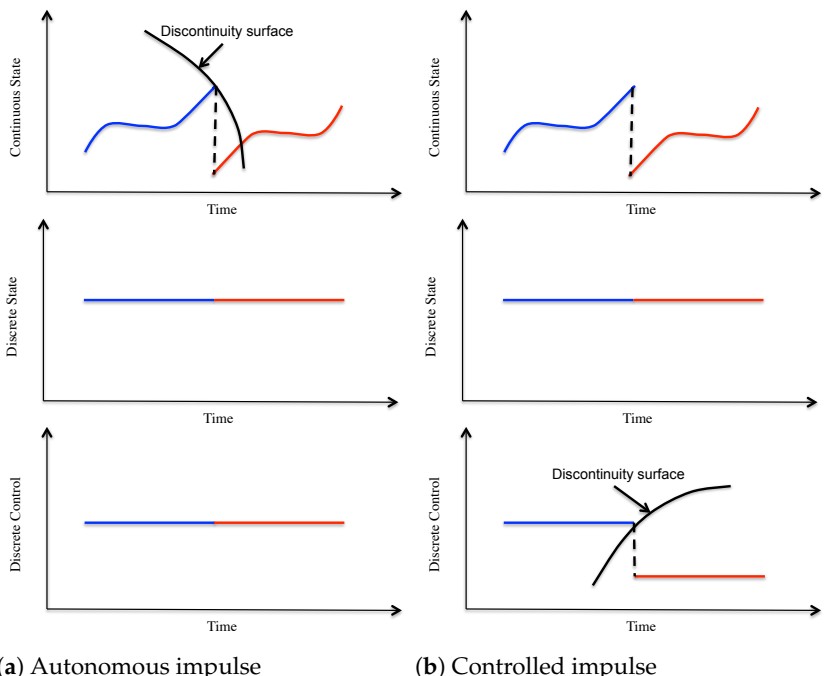

(**a**) Autonomous impulse      (**b**) Controlled impulse

**Figure 2.** Illustration of impulsive discrete events.

- Autonomous switching: An autonomous switching occurs when the continuous state trajectory crosses the discontinuity surface in the continuous state-time space (see Figure 1a). In this case, the discontinuity surface depends only on the continuous state and on time, i.e., $s = s(x, t)$. The switching causes the discrete state to change, whereas the continuous states before and after the switching are equal, i.e., $x(t_i^+) = x(t_i^-)$ and $q(t_i^+) = \phi(x, q, v, t_i^-)$. In the new discrete state, the continuous state trajectory follows different equation of motions than in the previous discrete state. In spacecraft systems, autonomous switching occurs, for example, when the electric engine is switched-off due to power availability constraints (e.g., the spacecraft crosses through the Earth-shadow or it is far from the Sun).

- Controlled switchings: Controlled switching differs from autonomous switching in that the discontinuity surface is not a function of the continuous state but it depends on the controls, i.e., $s = s(v, t)$. Therefore, the discrete event occurs in the control-time space (see Figure 1b). Controlled switching models logical decisions that can be made at a desired point of time to change the system dynamics, e.g., switching-off the electric engine for propellant savings reasons.

- Autonomous impulses: An autonomous impulse resets the value of the continuous state, when the continuous state trajectory hits the discontinuity surface (see Figure 2a). In a similar fashion than autonomous switching, the discontinuity surface depends only on the continuous state and on time, i.e., $s = s(x, t)$. However, after an autonomous impulse, the discrete state, and thus the differential equations, remains unchanged, whereas the continuous state jumps according to the transition maps function, i.e., $x(t_i^+) = \phi(x, q, v, t_i^-)$ and $q(t_i^+) = q(t_i^-)$. Examples for autonomous impulses in spacecraft dynamics are gravity assisted-maneuvers, since a discrete change is the heliocentric velocity is experienced when it encounters a planet in space and time.

- Controlled impulses: The difference of controlled impulses to autonomous ones is that the impulse is triggered by a discontinuity surface that depends on the controls, i.e., $s = s(v, t)$. Similarly to controlled switchings, the event occurs in the control-time space (see Figure 2b). Incrementing the velocity of a spacecraft by an instantaneous firing of a chemical engine is an example of a controlled impulse.

### 3.2. Problem Statement

The multiobjective HOCP is to find the set of feasible continuous $u(t)$ and discrete $v(t)$ control inputs belonging to the Optimal Pareto front that minimizes the multiobjective function $J(u, v, t)$, typically a vector-valued function of the hybrid system state, control and time:

$$\min J(u, v) = \mathcal{M} + \int_{t_0}^{t_f} \mathcal{L}(x, u, q, v, t) dt, \tag{3}$$

subject to Equations (1) and (2), and

$$u(t) \in \mathcal{U} \subset \mathbb{R}^{n_u}, \quad v(t) \in \mathcal{V} \subset \mathbb{Z}^{n_v}, \quad \forall t \in \left[t_0, t_f\right], \tag{4}$$

$$x(t) \in \mathcal{X} \subset \mathbb{R}^{n_x}, \quad q(t) \in \mathcal{Q} \subset \mathbb{Z}^{n_q}, \quad \forall t \in \left[t_0, t_f\right], \tag{5}$$

$$0 \le \mathrm{g}(x, u, q, v, t), \quad t \in \left[t_0, t_f\right], \tag{6}$$

$$x(t_0) = x_0(x, q, u, v, t_0), \qquad q(t_0) = q_0(x, q, u, v, t_0), \tag{7}$$

$$x(t_f) = x_f(x, q, u, v, t_f), \qquad q(t_f) = q_f(x, q, u, v, t_f). \tag{8}$$

In the above, the Lagrange integrand term $\mathcal{L} : \mathcal{X} \times \mathcal{Q} \times \mathcal{U} \times \mathcal{V} \times \mathbb{R} \longrightarrow \mathbb{R}^{n_j}$ is a vector real-valued function of the state and control variables and of time, and $n_j$ is the number of objective functions. The Mayer type part $\mathcal{M} : \mathcal{X} \times \mathcal{Q} \times \mathbb{R} \longrightarrow \mathbb{R}^{n_j}$ is a general vector function of the event transition times $t_i$ for $i = 0, \dots, N$ and of the continuous $x(t_i^-)$ and the discrete $q(t_i^-)$ states just before the discrete events and the continuous $x(t_i^+)$ and the discrete $q(t_i^+)$ states just after the discrete events. Thus, it is expressed as:

$$\mathcal{M} := \mathcal{M}\big(x(t_0^+), \dots, x(t_N^-); q(t_0^+), \dots, q(t_N^-); t_0, \dots, t_N\big). \tag{9}$$

Here, $t_0$ and $t_N = t_f$ are the beginning and final times, which are associated to an initial and final event function, respectively, whereas the remaining $N - 1$ transition times are related to interior event functions. The minimization of the multiobjective function in Equation (3) is subject to initial and terminal conditions on the state vector (7) and (8), admissible values for the continuous and discrete control and state variables (4) and (5) and further inequality constraints (6) given by the function $\mathrm{g} : \mathcal{X} \times \mathcal{Q} \times \mathcal{U} \times \mathcal{V} \times \mathbb{R} \longrightarrow \mathbb{R}^{n_g}$. Obviously, valid hybrid optimal trajectories must obey both the continuous and discrete dynamics. Let us define the optimal sequence of discrete events as:

$$\sigma = [(t_1, s_k), \dots, (t_i, s_j), \dots, (t_N, s_m)], \quad \text{for} \quad k, j, m \in 1, \dots, n_s \tag{10}$$

The key challenge when solving HOCPs is that the optimal sequence of discrete events $\sigma$ is not known a priori. Therefore, it has to be determined as part of the solution. Note that, in Equation (10) the sequence of discontinuity functions may have an arbitrary order, and even a discontinuity function can be activated more than once during the trajectory, unless otherwise specified, thus increasing the combinatorial complexity of the problem. Additionally, when facing multiobjective problems, instead of searching for a unique optimal law for the continuous and discrete control inputs as in single objective optimization, the aim is to obtain a whole set of different solutions that are equally optimal in terms of Pareto efficiency.

As an illustration, let us define the HOCP where a spacecraft is to travel from Earth to Saturn benefiting from as many gravity assisted maneuvers as desired and limited to a maximum time-of-flight. The patched conics approach is used and flybys are considered instantaneous, i.e., as discrete events. In such case, there are nine discontinuity functions, i.e., $(s_1, s_2, s_3, s_4, s_5, s_6, s_7, s_8, s_9)$ representing a planetary encounter with Mercury, Venus, Earth, Mars, Jupiter, Saturn, Neptune, Uranus, and Pluto, respectively. Multiobjective

solutions with respect to propellant mass and flight of time are to be obtained. In this case an optimal compromise sequence of gravity assists $\sigma_1$ is obtained, such that:

$$\sigma_1 = [(t_1, s_3), (t_2, s_2), (t_3, s_3), (t_4, s_5)] \tag{11}$$

where $t_1, t_2, t_3, t_4$ represent the optimal flyby maneuver times of the sequence Earth-Venus-Earth-Jupiter. A different compromise solution would result in a different optimal sequence.

## 4. Dynamical Modeling

The dynamical modeling of the problem requires to select a set of variables to represent the dynamical state of the system, to derive the set of dynamical differential equations to describe the evolution or time history of the state, and to choose the control variables, which represent the degrees of freedom of the system. In spacecraft trajectory optimization problems, the state of the vehicle is also referred as the trajectory (i.e., its position in space with respect to time), while the set of continuous/discrete differential equations are also known as Equations of Motion (EOM). The selection of these elements will mostly impact the speed, robustness, and accuracy of the resulting tool.

### 4.1. Continuous State Representation

The spacecraft is typically considered to be a point-mass. Thus, six independent parameters or generalized coordinates are necessary to describe its three-dimensional motion. In practice, there are several forms of representing the spacecraft state, each of them having positive and negative aspects [17]. They can be classified as sets based on position and velocity (e.g., cartesian or polar coordinates), and based on orbital elements (e.g., classical or equinoctial). An overview of the most prominent ones is presented hereafter:

- Cartesian State Vector (CSV): The most common model for describing a spacecraft trajectory refers to its position and velocity vectors. They are typically projected on an inertial Cartesian frame, such that $x_{CSV} = [r_x, r_y, r_z, v_x, v_y, v_z]$. Here, $(r_x, r_y, r_z)$ and $(v_x, v_y, v_z)$ are the projections of the position $\mathbf{r} \in \mathbb{R}^3$ vector, and of the velocity vector $\mathbf{v} \in \mathbb{R}^3$, respectively.
- Polar State Vector (PSV): They are mainly used for two-dimensional or planar representations of the problem dynamics. They consists on the following set: $x_{PSV} = (r, \theta, v, \psi)$, where r is the distance to the central body, $\theta$ is the polar angle, v is the modulus of the velocity with respect to an inertial frame, and $\psi$ is the flight path angle.
- Classical Orbital Elements (COE): Another form of mathematical model to represent the spacecraft dynamics is in terms of classical orbit elements $x_{COE} = (a, e, i, \Omega, \omega, M)$. They are named as the semimajor axis, eccentricity, inclination, right-ascension of the ascending node, argument of perigee, and mean anomaly, respectively. Instead of the true anomaly, the mean motion, the true anomaly or the eccentric anomaly can be used [18].
- Modified Equinoctial Elements (MEE): The other model for completely defining the state of the spacecraft is by the use of the set of modified equinoctial orbital elements $x_{MEE} = (p, f, g, h, k, L)$. Here, $p$ is the semilatus rectum and $L$ is named the true longitude. The elements $(f, g)$ are related to the projection of the eccentricity vector on the inertial frame, while $(h, k)$ are associated to the inclination of the orbit.

The CSV representation is widely used for low-thrust interplanetary trajectories. They allow to naturally impose the restrictions associated to flyby or rendezvous a planet, as well as to easily formulate the problem including multibody gravitational attractions. Additionally, the resulting cartesian EOM are singularity free. However, in planetocentric environments, where multirevolution occurs, strong oscillations of the cartesian state variables occurs, which decreases robustness. Thus, more efficient state representations are required to reduce the computational cost for these transfers. The PSV formulation is simple but is rather inflexible, as it is limited to planar transfers. This fact may not be

a problem during the preliminary design of interplanetary transfers, since most planets almost lie in the same orbital plane.

Conversely, the COE representation is typically applied in planetocentric environments because the trajectory can be integrated faster than with CSV for the same accuracy. They are intuitive as they are related to the physical geometry of the trajectory. For low-thrust trajectories this formulation is appealing because the solution can be described in terms of "almost constant" orbital elements. This fact has allowed many authors to obtain analytical or semianalytical representations of the trajectory, which speed up the computation of the trajectory. Unfortunately, they have a number of singularities that may complicate the numerical integration. For instance, at zero inclination ($i = 0$) the right ascension of ascending node ($\Omega$) loses meaning. Similarly, for zero eccentricity ($e = 0$) the argument of perigee ($\omega$) becomes undetermined. This is the case of many of the orbits of interest such as GEO. These singularities cause rapid oscillations when the spacecraft is near a singular point [19].

Similarly, the MEE is used for multirevolution transfer in planetocentric environments. They are nonsingular for all values of eccentricity and inclination, increasing robustness. Therefore, they are most used in low-thrust orbit raising transfers to GEO. However, unlike COE, the physical interpretation of the MEE set is not intuitive. Both COE and MEE allow to easily imposed the constraint of reaching a certain orbit, where the specific location in the orbit is not important. They also permit to fasten the integration of the EOM by applying averaging techniques. However, neither COE or MEE are well suited when perturbations of the two-body problem are significant, such as transfers to the moon or to libration points.

There is significant freedom in the choice of a suitable set of state variables or orbital elements. Therefore, depending on the specific mission or on the mission designer's experience, one set may be used in favor of others to provide better results in terms of speed, accuracy, and robustness. Notably, other forms of state representations than the ones explained herein may be used for spacecraft trajectory optimization. To be more specific, there are twenty two identified candidate orbit element sets plus variations. These other forms of orbital elements are well explained in a survey presented by Hintz [17]. Additionally, the evolution of the spacecraft mass $m$ is typically required to fully describe the dynamics of the system. It is used to compute the acceleration $a_T$ produced by the spacecraft given the thrust force $T$ produced by the low-thrust propulsion subsystem, and it varies with respect to time as propellant mass is consumed.

*4.2. Continuous Controls*

A low-thrust engine is usually controlled by continuously varying the direction and modulus of the acceleration produced by the engine. The engine acceleration $\mathbf{a}_T$ can be expressed as a function of the thrust generated, which generally depends on the spacecraft relative position with respect to the Sun, the total mass, and the continuous control variables as follows:

$$\mathbf{a}_T = \frac{T}{m}\boldsymbol{u}(t) \tag{12}$$

In the above, $\boldsymbol{u}(t) = [u_1(t), u_2(t), u_3(t)]$ represent the direction cosines of the thrust pointing vector with respect to an inertial reference frame. The following path constraint have to be fulfilled $\sqrt{u_1^2 + u_2^2 + u_3^2} = 1$. Alternatively, the thrust azimuth $\alpha$, and declination $\beta$ steering angles can be considered as control variables. In such case, the dimension of the control space is reduced from three to two, i.e., $\boldsymbol{u}(t) = [\alpha(t), \beta(t)]$, and the path constraint do not need to be applied. The thrust acceleration vector can be computed as:

$$\mathbf{a}_T = \frac{T}{m}[\cos\alpha\cos\beta, \sin\alpha\cos\beta, \sin\beta] \tag{13}$$

This approach is flexible, since it allows to represent the continuous control for all the possible scenarios in low-thrust trajectory optimization problems. However, having to determine the thrusting angle at every time instant results in a time-consuming process.

Thus, a different selection of control variables are possible to obtain difference performances. Notably, during the preliminary design of spacecraft trajectories, it is common to use predefined or heuristic control laws, such that the thrust direction is prescribed as a function of a small set of static controls or parameters. Heuristic control laws generally yield suboptimal trajectories, but follow a policy that a mission designer deems acceptable for the preliminary design. Some predefined control laws may allow to obtain an analytical representation of the trajectory. They can be categorized into six main groups, depending on the heuristic function that is used to parametrize the control:

- Blended Control (BC): The optimal thrust steering that maximize the variation (i.e., increase or decrease) of a set of orbital elemenst element independently or each other, $\boldsymbol{u}_x(\boldsymbol{x}) \in \mathbb{R}^{n_x}$ are computed as a function of the position in the orbit. They are commonly obtained analytically. Then, the complete control law to simultaneously modify all the elements of the state vector results from the following weighted sum:

$$\boldsymbol{u}^*(W_x, t) = \sum G_x(t)\boldsymbol{u}_x(\boldsymbol{x}) \tag{14}$$

  where $W_x \in \mathbb{R}^{n_x}$ are time-varying or static weighting functions that fulfills $\sum W_x(t) = 1$. Their time-discretized values $W_x(t_i)$ are the unknowns to be determined. Commonly, BC-based methods are derived for MEE or COE formulations, and allow to naturally reach the target orbit, avoiding the need to impose final boundary constraints.This type of control law is rather used in planetocentric environments, where the rendezvous with a target true anomaly may not be required.
- Calculus of Variations based (COV) The Pontryagin Minimum Principle (PMP) [20] is used to obtain the optimal control history. For a minimum-time continuous optimal control problem, the optimal thrust direction will have the following form:

$$\boldsymbol{u}^*(\lambda, t) = -\frac{M(\boldsymbol{x})\lambda(t)}{||M(\boldsymbol{x})\lambda(t)||} \tag{15}$$

  where $M(\boldsymbol{x})$ is state-dependent matrix resulting from solving the PMP. Here, $\lambda(t)$ are known as the costates, and represent the new continuous controls. Unfortunately, this approach lack of flexibility since an analytical reformulation of $M(\boldsymbol{x})$ is required every time a new constrained is added or a new-objective function is considered. Beside, if the problem combines hybrid dynamics, the formulation of this control law becomes much more challenging.
- Lyapunov Control (LC): It defines an energy-like (i.e., a positive-definite) scalar Lyapunov function of the state $V(\Delta x(t), W_x) \in \mathbb{R}$. Here, $\Delta x(t) = \boldsymbol{x}(t) - \boldsymbol{x}_f$, and $\boldsymbol{x}_f$ is the target state. The set of constant parameters or static controls $W_x \in \mathbb{R}^{n_x}$ are to be determined as part of the solution. The Lyapunov function has to fulfill the following condition:

$$\dot{V}(W_x) = \nabla_x V(\Delta x, W_x) \cdot \mathrm{f}(\boldsymbol{x}, \boldsymbol{u}) \leq 0 \tag{16}$$

  The thrust steering law is then obtained by minimizing the variation of $\dot{V}$ with respect to the control law (i.e., making it as negative as possible) as follows:

$$\boldsymbol{u}^*(\boldsymbol{z}, t) = \arg\min_u \nabla_x V(\Delta x(t), W_x) \cdot \mathrm{f}(\boldsymbol{x}, \boldsymbol{u}) \tag{17}$$

  Notably, this control law naturally drives the spacecraft to the desired final state, avoiding the need to include the final boundary conditions in the problem.
- Shape-based Approaches (SB): In this approach, the state vector $\boldsymbol{x}(t)$, usually the trajectory, is assumed to have a predefined form, e.g., $\boldsymbol{x} = \boldsymbol{x}(\boldsymbol{z}, t)$, where $\boldsymbol{z} \in \mathbb{R}^{n_z}$ are

the set of parameters to be determined. The control law is obtained by forcing the EOM to be satisfied:

$$u^*(z,t) : \dot{x}(z,t) - f(x(z,t), u^*, t) = 0 \tag{18}$$

An analytical solution for the control is derived therefrom. Note that the obtained control may not satisfy the constrained related to the maximum thrust available. Thus it may lead to unfeasible trajectories. The solution may not fulfill the boundary constraints, thus they must be included as part of the problem.

- Neurocontroller (NC): The problem of finding an optimal strategy that leads to an optimal trajectory is thus transformed into the determination of the optimal network transfer function $N : \mathcal{X} \times \mathbb{R}^{n_z} \times \mathbb{R} \longrightarrow \mathcal{U}$. This function acts as a map from the current spacecraft state $x$, the desired final state $x_f$, and the network's internal parameters $z \in \mathbb{R}^{n_z}$ to the instantaneous steering. Thus, it holds that:

$$u^*(z,t) = N(z, x_f, x, t) \tag{19}$$

The controller parameters $z \in \mathbb{R}^{n_z}$ are to be determined as part of the solution.

- Finite Fourier Series (FFS). The low-thrust steering history is assumed to be represented by a Finite Fourier series expansion, such that:

$$u^*(a_k, b_k, t) = \sum_{k=0} a_k(t) \cos\left(\frac{2\pi k\theta}{\Delta\theta}\right) + b_k(t) \cos\left(\frac{2\pi k\theta}{\Delta\theta}\right) \tag{20}$$

where the time-varying or static coefficients $a_k$ and $b_k$ are the continuous controls. The angle $\theta$ represents the orbit anomaly, and $\Delta\theta$ represents the with of the interval in which the Fourier expansion applies. Note that, increasing the number of coefficients will improve the accuracy of the representation at the cost of increasing the number of unknowns and the complexity.

### 4.3. Discrete States

Up to this point, defined state and control variables have been classified as continuous-valued, i.e., they can assume infinite values in a given continuum. However, for certain problems, it is interesting to include discrete-valued or discontinuous states, i.e., they can take values in a finite or countable set. For example, a discrete state variable $q \in \mathbb{Z} \in \{0, 1\}$ can describe the different working modes of an electric engine (*on* or *off*). When switched-on ($q = 1$), the engine operates at maximum thrust, whereas when switched-off ($q = 0$) the thruster is coasting. In such case, the acceleration produced by the electric engine can be formulated as follows:

$$a_p : \begin{cases} a_p = \dfrac{T}{m}\, \mathbf{d}, & \text{if} \quad q = 1 \\ a_p = 0 & \text{if} \quad q = 0 \end{cases} \tag{21}$$

Note that changing the mode of operation implies changing the set of differential equations. Therefore, the time-history of the discrete state is required to determine the trajectory. Additionally, the discrete states may be used to model discrete sets containing available options for the design of the mission architecture, e.g., different launcher options, different propulsion systems with different operational points, or different attitude subsystems. Each alternative provides distinct performances, and consequently a different resulting trajectory. The optimal solution will contain the optimal set of discrete states, e.g., the optimal launcher, propulsion and attitude system. This feature is a key requirement deriving from concurrent engineering principles.

*4.4. Discrete Controls*

Discrete controls may be included in the system to model controller switchings or changing operating modes. The switch between modes of operation can be managed by a binary control input $v(t) \in \mathbb{Z} \in \{0,1\}$. For instance, a coasting state is required when $v = 0$, while a thrusting state is required with $v = 1$. Furthermore, other controlled decisions can be modeled, such as performing chemical maneuvers, changing between the thruster operational modes (i.e., different thrust and specific impulse values), or starting data downlink with a ground station.

*4.5. Continuous Dynamics*

The formulation chosen to represent the continious dynamics highly impacts the computational speed as well the solution accuracy. Let consider a spacecraft traveling in space under the gravitational attraction of *n*-bodies in the solar system and subject to the acceleration produced by a low-thrust engine and other space environmental effects (see Figure 3). Its continuous dynamics can be generally described as a Perturbed Restricted N-Body Problem (PR-NBP). In case the gravitational bodies are perfectly spherical, the PR-NBP is mathematically expressed in CSV coordinates with respect to an inertial cartesian reference frame as follows:

$$\dot{\mathbf{v}} = -\sum_{i=1}^{n} \frac{\mu_i(\mathbf{r}(t) - \mathbf{r}_i(t))}{|\mathbf{r}(t) - \mathbf{r}_i(t)|^3} + \mathbf{a}_T + \mathbf{a}_P, \quad \dot{\mathbf{r}} = \mathbf{v}, \quad \dot{m} = \dot{m}(\mathbf{x}, \mathbf{u}, t) \tag{22}$$

Here, $\mu_i$ and $\mathbf{r}_i$ are the gravitational constant and position vector of the $i^{th}$ attracting central mass, respectively, whereas $\dot{m}$ is the propellant consumption rate of the propulsion system. Note that if $n = 2$ or $n = 1$ the formulation is known as the perturbed-restricted three-body-problem (PR-3BP) or as the perturbed-restricted two-body-problem (PR-TBP), respectively. The perturbing acceleration $\mathbf{a}_P$ represents the summation of any accelerations due to the space environment other than the gravitational attraction (e.g., solar radiation, atmospheric drag). The EOM (Equation (22)) can be formulated using other state vector such as PSV, COE or MEE.

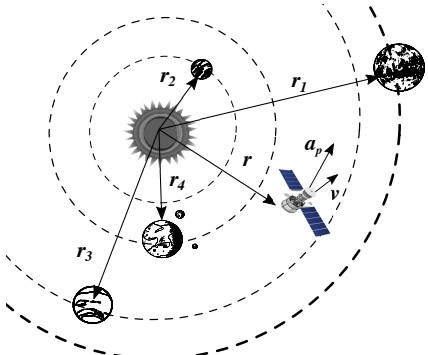

**Figure 3.** Perturbed Restricted N-body Problem Illustration.

Computing trajectories under the PR-NBP formulation, yet highly accurate and required for the detailed design, is computationally expensive. Thus, simplified or surrogate models are demanded for the preliminary design. The first step is to reduce the number of attracting bodies up to an acceptable value. For instance, a low-thrust mission to the Moon requires a PR-3BP formulation. However, PR-TBP dynamics provides suitable results for transfers between Earth-orbits. Notably, for interplanetary transfers, a patched-conic approach is often assumed. This simplification splits the trajectory into a sequence PR-TBP, i.e., the trajectory changes from being heliocentric to planetocentric when the spacecraft enters the sphere of influence of a particular planetary body. An additional approximation assumes that the radius of this sphere is infinitesimal and the flyby oc-

curs instantaneously [21]. As a second step, analytical solutions, averaging techniques or asymptotic analysis can be applied to further speed-up the process at the cost of fidelity.

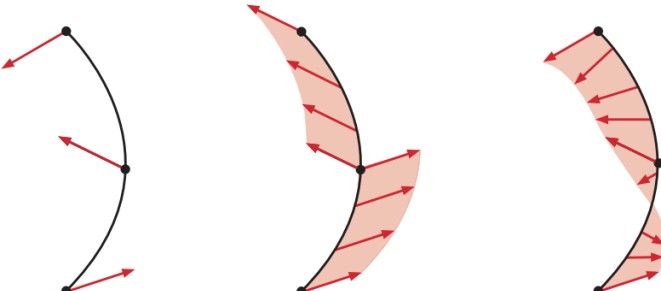

**Figure 4.** From left to right: The Kepler model, the Stark model, and the Continuous model.

- Analytical solutions: Analytical techniques were at the origin of spacecraft trajectory optimization. They seek to obtain closed-forms solutions for the dynamical systems, such that the EOM do not need to be integrated.

$$\dot{x} = \mathrm{f}(x, u, t) \longrightarrow x = x(x, u, t) \tag{23}$$

These techniques are only available for special cases. Two well-known and widely used analytical solutions are the Kepler and Stark models. A graphical representation of these techniques along with the continuous model is represented in Fig. **??**.

  - Kepler Model (KM): It is a reduced model that uses pure Keplerian arcs connected at nodes with impulsive velocity vector discontinuities that approximate the effect of performing a low-thrust maneuver during the Keplerian arc.
  - Stark Model (SM): The Stark model yields exact closed-form solutions for a spacecraft in a two-body gravitational field subject to a thrust acceleration that is inertially constant in both magnitude and direction.

  Additionally, analytical solutions can be derived under constant radial or tangential thrust without space perturbations, even including some environmental effects, such as the Earth oblateness.
- Asymptotic solutions: The propulsive acceleration is considered as a perturbation effect acting on a well-known or unperturbed trajectory (e.g., a Keplerian orbit). Thus, the perturbed trajectory can be approximated as a series expansion:

$$\dot{x} = \mathrm{f}(x, u, t) \longrightarrow x(\epsilon, t) \approx x_0(t) + \epsilon x_1(t) + \mathcal{O}(\epsilon^2) \tag{24}$$

where $\epsilon$ is a nondimensional thrust acceleration, and has to fulfill that $\epsilon \ll 1$, $x_0$ is the unperturbed trajectory, and $x_1$ is the first-order perturbation term, which can be obtained analytically under certain circumstances (e.g., constant tangential or radial acceleration). Commonly, second-order terms are not included in the expansion.
- Averaging techniques: The method of averaging consists in the elimination of high-frequency components from the EOM by averaging over a short time scale (typically the orbital period). The averaged equations contains only secular and long-periodic terms.

$$\dot{x} = \mathrm{f}(x, u, t) \longrightarrow \dot{\bar{x}} = \frac{1}{\mathrm{T}} \int_t^{t+\mathrm{T}} \mathrm{f}(x(t), u(t), t)dt \tag{25}$$

where $\bar{x}$ is the mean state vector, and T is the orbital period. This is particularly useful in planetocentric scenarios with multiple-revolutions due to the quasi-periodic nature of the orbits. However, averaging results in a loss of exact position information which may be desired to assess the power availability to the spacecraft or to rendezvous with a celestial body.

### 4.6. Discrete Dynamics

The spacecraft dynamics presented in Equation (22) are continuous, since they are governed by differential equations. However, spacecraft dynamics may include discrete-event dynamics. Discrete events produces instantaneous changes in the spacecraft continuous or discrete state. Performing a gravity assist maneuver or switching on/off the electric engine are examples of discrete events. Notably, the sequence and number of discrete-events, i.e., the sequence and number of flybys or electric engine switchings, is not known a-priori.

#### 4.6.1. Flybys

Let us define the continuous state vector of a planet $b_j$ as $\boldsymbol{x}_{b,j}(t) = [\mathbf{r}_{b,j}, \mathbf{v}_{b,j}]$, where $\mathbf{r}_{b,j}(t) \in \mathbb{R}^3$ and $\mathbf{v}_{b,j}(t) \in \mathbb{R}^3$ represent its position and velocity heliocentric vectors, respectively. Flybys are assumed to produce an instantaneous change in the heliocentric velocity of the spacecraft, given by the transition map $\phi_{fb,j}$ and occurring when the position vector of the spacecraft intersects the discontinuity surface $s_{fb}$:

$$\text{Flybys}: \begin{cases} \phi_{fb,j}: \mathbf{v}(t_i^+) = \boldsymbol{\Delta}_j(\mathbf{v}(t_i^-), \boldsymbol{p}_j), & q(t_i^+) = q(t_i^-), & \boldsymbol{r}(t_i^+) = \boldsymbol{r}(t_i^-) \\ s_{fb,j}: ||\mathbf{r} - \mathbf{r}_{b,j}(t)|| = 0, & j \in \{1, \ldots, n_s\} \end{cases} \quad (26)$$

As modeled by the discrete event function $s_{fb,j}$, a flyby is only possible if the spacecraft heliocentric position matches the heliocentric position of a planet. Note that there are as many discontinuity surfaces as $n_{fb}$ available planets to flyby. Following the aforementioned approach, if a planet $b_j$ is encountered at $t_i$, the heliocentric post-flyby velocity $\mathbf{v}(t_i^+)$ can be obtained assuming a hyperbolic trajectory around the planet, which is a function of the preflyby velocity $\mathbf{v}(t_i^-)$, the planet heliocentric velocity $\mathbf{v}_{b,j}(t_i^-)$ and additional static control parameters $\boldsymbol{p}_j = [r_{p,j}, \zeta_j]$, which are subject to optimization. The additional parameters are the minimum distance of approach $r_{p,j}$ and the B-Plane angle $\zeta$. Check reference [22] for further details.

#### 4.6.2. Engine on-off Switchings

The switch between the thrust/coast modes of operation can be described by a controlled discrete event or by an autonomous event. The former occurs as a consequence of a controlled decision, for propellant savings reasons, whereas the latter occurs as a consequence of the power subsystem requirements (when there is not enough power available for the engine to operate). Both are summarized in the following functions:

$$\text{Switching-on}: \begin{cases} \phi_{on}: q(t_i^+) = 1, & \mathbf{v}(t_i^+) = \mathbf{v}(t_i^-), & \boldsymbol{r}(t_i^+) = \boldsymbol{r}(t_i^-) \\ s_{on}: q(t_i^-) = 0, & v(t_i^-) = 1, & 0 \leq g(\boldsymbol{x}, q, \boldsymbol{u}, \boldsymbol{v}, t_i^-) \end{cases} \quad (27)$$

$$\text{Switching-off}: \begin{cases} \phi_{off}: q(t_i^+) = 0, & \mathbf{v}(t_i^+) = \mathbf{v}(t_i^-), & \boldsymbol{r}(t_i^+) = \boldsymbol{r}(t_i^-) \\ s_{off,1}: q(t_i^-) = 1, & v(t_i^-) = 0 \\ s_{off,2}: q(t_i^-) = 1, & 0 \geq g(\boldsymbol{x}, q, \boldsymbol{u}, \boldsymbol{v}, t_i^-) \end{cases} \quad (28)$$

Here the event surface $s_{on}$ refers to the controlled switching-on whereas $s_{off,1}$ and $s_{off,2}$ represents the event surface for the controlled and autonomous switching-off, respectively. The function $g$ imposes the constraint related to the power system, i.e., when $g > 0$ there is not enough power available and the thruster cannot operate. Notably, the engine on-off switchings can be parametrized to reduce the complexity of the problem. For example, a coasting mechanisms based on the effectivity of the maneuver, i.e., as a function of the instantaneous rate of change of an orbital element with respect to the maximum obtainable. If this efficiency factor is below a threshold, the spacecraft turns to coasting mode, until the efficiency improves. This strategy is typically combined with BC and LC continuous control parametrization.

## 5. Objective Functions

The objective function, also called value function or performance index, represents the cost of the mission in minimization problems or the benefit in maximization ones. The form defined in Equation (3) is known as the Bolza objective function [13]. Various forms of objectives can be categorized with respect to two different aspects: the type and number of objectives. In most trajectory optimization problems, according to Conway [14], there are two common types of objectives: either some function related to the control effort or to the time required to accomplish the mission. The former typically relates to the spacecraft thrust acceleration level, $J = \int_{t_0}^{t_f} |\mathbf{a}_T| dt$, or to the propellant mass consumed, $J = m(t_f) - m(t_0)$. The latter simply takes the Mayer form $J = t_f$. Alternative objectives, such as launch mass or absorbed radiation during the passage through the Van-Allen belts, as well as mission-specific criteria may be considered. Regarding the number of objective functions $n_k$, the problem can be classified as either single-objective or multiple-objective.

- Single-objective: The goal is to search for a solution in the feasible set that provides the minimum value of a scalar-valued function, i.e., $n_j = 1$. In this case, a single-point solution, under mild regularity assumptions, is obtained. From a mathematical point of view, a feasible solution $(u^*, v^*)$ is optimal if it satisfies the following condition:

$$J(u^*, v^*) \leq J(u, v), \quad \forall u \in \mathcal{U} \quad \text{and} \quad \forall v \in \mathcal{V} \tag{29}$$

- Multiobjective: The aim is to minimize a vector-valued function formed by $n_j > 1$ conflicting criteria, i.e., $J = [J_1, J_2, \ldots, J_{n_j}]$. The solution in the objective space typically consists of a $(n_j - 1)$-dimensional hypersurface [23] known as the Pareto-optimal set (Pareto-optimal set is also known as Pareto front, Pareto frontier, Pareto-efficient set or nondominated front.) [24]. A feasible solution $(u^*, v^*)$ is weak Pareto-optimal if there does not exit another feasible solution $(u, v)$ that could improve all the objectives simultaneously such that:

$$J_i(u, v) \leq J_i(u^*, v^*), \quad \forall i \in \{1, \ldots, n_j\} \quad \forall u \in \mathcal{U} \quad \text{and} \quad \forall v \in \mathcal{V} \tag{30}$$

Otherwise, the point $(u^*, v^*)$ is said to be dominated.

Solving multiobjective optimization problems, also known as vector optimization or multipurpose optimization, is far more difficult and computationally expensive than solving single-objective problems. However, mission-planning during the preliminary design greatly benefits from the trade-offs provided by multiobjective optimization. In fact, most optimization problems in low-thrust trajectory design have multiple objectives that are often equally important and conflicting. Thus, any concurrent engineering study must involve a multiobjective optimization process. As an example, consider the optimization of propellant mass consumed and transfer time-of-flight. The feasible objective space along with six designs is illustrated in Figure 5. Because both propellant mass and flight time are minimized, the Pareto front is located in the lower left region of the feasible objective space. Design-1, design-2 and design-3, are along the Pareto front and compose the Pareto-optimal; all other designs are nonoptimal. Although design-3 has the lowest propellant mass, design-1 has shorter time of flight; thus they are equally optimal in terms of Pareto. Note that, solution design-1 would have been obtained by a single-objective problem minimizing time-of flight. Similarly, solution design-3 would be the solution of uniquely minimizing propellant mass. Solution design-2 could be obtained by minimizing a scalar combination of time of-light and propellant mass. For further background in the associated multiobjective optimization in engineering applications, the reader should refer to Marler et al. [25].

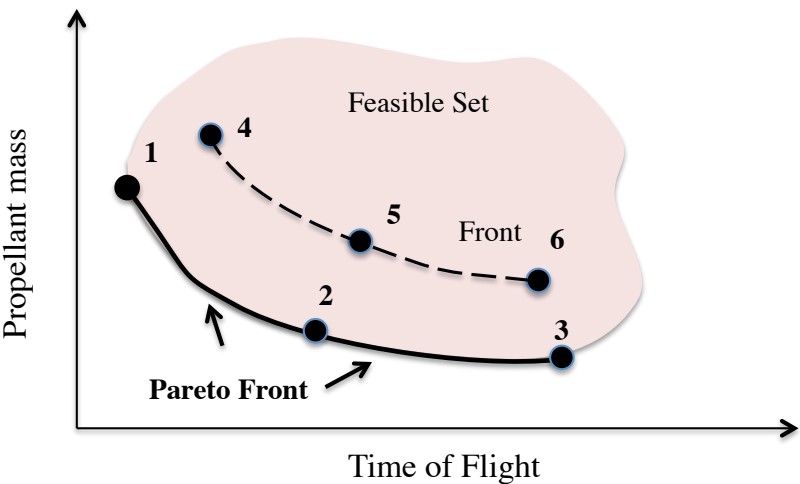

**Figure 5.** Illustration of a Pareto front.

## 6. Approaches and Solutions for COCPs

Hitherto, the elements required to properly formulate a spacecraft trajectory optimization problem have been presented, namely objective functions, continuous and discrete spacecraft state representation, continuous and discrete dynamics, and continuous and discrete control variables. The next step is to select a proper approach for finding the optimal solution. The chosen solution approach will impact the flexibility, robustness, optimality and automation of the method. Historically, low-thrust trajectory optimization problems have been formulated as purely continuous optimal control problems (COCP). Notably, well-developed techniques for solving COCP are totally or partially transferred to solve more complex trajectory optimization problems such as HOCP. Therefore, in this section, the different solution methods presented in the literature for solving COCP, will be characterized. For the sake of brevity, only an overview of approaches with a brief discussion is provided herein. For a fundamental background on the associated methodologies, the reader should refer to [12–14].

As a rule, two types of approaches exists: analytical and numerical approaches. Analytical approaches produce closed-form solutions for the optimal trajectory. Since they can only be obtained in special cases, they are seldom feasible for most spacecraft trajectory optimization problems. The majority of researchers have been dedicated to numerical methods in order to solve more meaningful problems. Numerical approaches can be divided in three well-known methods: indirect methods, direct methods and dynamic programming. Indirect methods rely on the Pontryagin minimum principle (PMP), dynamic programming on the Hamilton-Jacobi-Bellman theory, and direct methods on the Karush-Kuhn-Tucker (KKT) optimality conditions. Furthermore, each method result in a different mathematical problem that can be solved with the aid of gradient-based, heuristic or hybrid techniques. Each combination exhibits differentiating positive and negative aspects. Hereafter, an overview of these approaches along with their related techniques will be briefly discussed. The overall schema of numerical approaches is depicted in Figure 6.

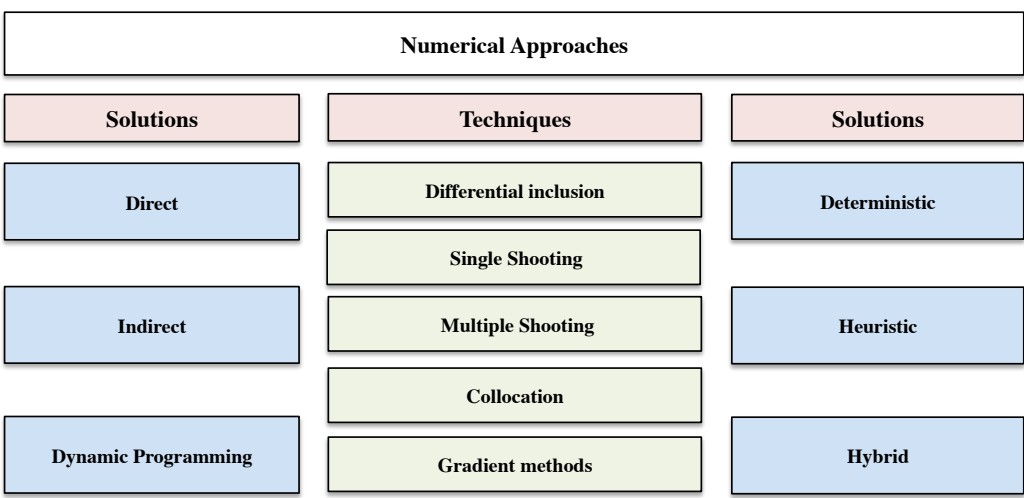

**Figure 6.** Numerical approaches, techniques and solutions for COCPs.

### 6.1. Indirect, Direct, and Dynamic Programming Approaches

In the following lines, an overview of the three most important categories regarding low-thrust trajectory optimization is presented:

- Indirect Approach: In the indirect approach, the goal is to solve the multipoint boundary value problem (MPBVP) that results from applying the PMP [20]. The PMP characterizes the first-order necessary conditions that an optimal solution must satisfy. its derivation involves the determination of the states and costates, which must obey the Euler-Lagrange equation. Notably, the minimum principle allow to obtain the continuous control as a function of the state and costate at each instant, explicitly or numerically . Furthermore, a set of additional constraints, namely transversality, and complementary conditions, must be satisfied [26].

- Direct Approach: The basic idea of direct methods is to transcribe the COCP into a nonlinear programming problem (NLP), where the objective function (Equation (3)) is "directly" optimized. The transcription process requires the discretization of the control variables in a time-grid. The goal of a NLP problem is to determine a vector of unknown decision variables that comply with a set of nonlinear constraints, including equality and inequality restrictions. An optimal solution to the NLP problem has to fulfill first-order necessary optimality conditions. These conditions are known as the Karush-Kuhn-Tucker conditions (KKT) [27,28]. The NLP is then numerically solved using well-known optimization techniques [13].

- Dynamic Programming Approach: The method of Dynamic Programming is based on the Bellman's principle of optimality [29]: "An optimal policy has the property that whatever the initial state and initial decision are, the remaining decisions must constitute an optimal policy with regard to the state resulting from the first decision." Even though Dynamic Programming was originally developed for discrete-time systems, it was extended to continuous-time problems. The continuous-time equivalent of the Bellman's principle resulted in the Hamilton-Jacobi-Bellman (HJB) theorem [30]. In this case, a set of partial differential equations must be solved first.

Moreover, indirect and direct methods typically involve one of the following techniques to impose the dynamical equations in the solution:

- Single shooting: The trajectory is integrated using time-marching methods from $t_0$ upon reaching the final time $t_f$. In this case, the initial state (and costates) are unknowns to be determined, and boundary constraints are imposed at the end of the integration.

- Multiple shooting: The time interval $[t_0, t_f]$ is broken up into $N + 1$ subintervals. The trajectory is integrated over each subinterval $[t_i, t_{i+1}]$ with the initial values of the state (and adjoints) at each subinterval being unknowns that need to be deter-

mined. Additionally, continuity conditions have to be enforced at the interface of each subinterval.
- Collocation: The states (and costates) are discretized over a predefined time-grid, such that they are known only at discrete points. The system-governing equations are transformed into discrete defect constraints, which relate the values at the beginning of the subinterval to the values at the end. Different methods are characterized by the choice of quadrature rule to approximate the differential equations between each two subintervals: local and global collocation methods.

Indirect methods can also be solved with gradient techniques, that involve a forward integration of the dynamical system and a backwards integration of the adjoint equations. To perform the forward integration and the initialization of the adjoint variables, a control function of the time has to be initially guessed. These unknowns are the decision variables, which are iteratively varied until the optimality conditions are satisfied. Depending on the update procedure for the control, gradient methods of first order [31] and second order [32] are distinguished. A different technique that lies within the direct methods category is differential inclusion. In this technique, only the state variables are discretized over a pre-defined time-grid. The equations of motion are enforced at each discrete time by applying inequality constraints on the rates of change of the states. These inequality constraints are obtained by substituting the upper and lower bounds on the control vector into the equations of motion. Regarding dynamic programming, the most successful technique relies in Differential Dynamic Programming(DDP) [33]. It is a gradient-based second-order technique that relies on HJB theorem and successive minimization of quadratic approximations of the problem DDP proceeds by iteratively performing a backward pass on the nominal trajectory to generate a new control sequence, and then a forward pass to compute and evaluate the cost of the trajectory.

*6.2. Gradient-Based, Heuristic, and Hybrid Solutions*

Most previous approaches (e.g., indirect/direct single/multiple shooting and collocation) have converted the COCP to the problem of determining an unknown vector of decision variable. For direct methods, the unknown decision vector has to fulfill a set of nonlinear constraints, while minimizing an objective function (i.e., solving an NLP problem). On the other hand, in indirect methods the unknown parameters have to meet a set of nonlinear constraints (i.e., solving a MPBVP). Methods for solving NLPs and MPBVP can be classified as gradient-based (also known as deterministic methods) heuristic or hybrid algorithms. They all are iterative methods that use a different set of rules for evolving. Hereafter, the main lines for each of them are drawn:

- Gradient-based: In a gradient-based method, an initial guess is made of the unknown decision vector $z$. At the $k^{th}$ iteration, a search direction $p_k$, and a step length $\alpha_k$, are determined. The search direction provides a direction in $\mathbb{R}^{n_z}$ along which to change the current value $z_k$, while the step length provides the magnitude of the change. The update from $z_k$ to $z_{k+1}$ has the form: $z_{k+1} = z_k + \alpha_k p_k$. The iterations proceed until the KKT conditions are met. To compute the search direction, these methods require the user provide information for the gradient of the constraint and the objective function (if necessary). The most widely used methods are classified as sequential quadratic problems (e.g., SNOPT, NPSOL) or interior point methods (e.g., IPOPT, KNITRO). Extensive information about their implementations can be found in Refs. [34,35], respectively.
- Heuristic: The search is performed in a stochastic/metaheuristic manner without requiring gradient information. The most known class of heuristics are evolutionary algorithms. They start by generating a set of candidate solutions or individuals $z_{i,0}$ for $i = 1, \ldots, n$, termed population. Thereafter, the population is iteratively modified by applying a set of stochastic rules $\Pi : \mathcal{Z} \longrightarrow \mathcal{Z}$, which may incorporate random processes, such that the population at $(k + 1)^{th}$ iteration is computed as $z_{i,k+1} = \Pi(z_{i,k})$, and the iterations proceed until a stopping criteria is met (e.g.,

max number of iterations). The candidate with the lowest cost is deemed as the solution to the problem. Well known stochastic rules are genetic algorithms (GA) [36], which emulate evolutionary processes in genetics, and particle swarm optimization (PSO) [37], which is based on the idea of swarms of animals.

- Hybrid: Hybrid approaches combine a set of rules exploiting gradient-information and a set of rules based on heuristics searches to iteratively operate over a solution or a set of candidate solutions. Gradient-information is exploited to drive the constraints to zero, while heuristic rules are applied to efficiently explore large design domains or to manage integer variables. They are typically combined on a two-loop approach. The heuristic solver operates over a subset of decision variables in the outer loop. In the inner loop, the remaining subset of design parameters are optimized with the gradient-based method.

*6.3. Discussion*

The main benefit of using the indirect approach is that it provides assurances that the first-order optimality conditions are satisfied. Additionally, they may offer an interesting theoretical insight into the problem physical and mathematical characteristics. However, they are not flexible, since explicit derivations of the costate and control equations are required, which can be difficult depending on the OCP being considered. Numerical techniques applied to the resulting MPBVP normally require an appropriate initial guess of the costates, which is often nonintuitive since they generally do not have physical interpretations. Moreover, they are not robust, since the resulting trajectory is sensitive to the values of the costates. The indirect approach is further complicated by the need to reformulate the MPBVP when different state variables, constraints and dynamics are considered. Because of these practical difficulties, indirect methods are not suitable to solve highly constrained spacecraft trajectory optimization problems, nor problems where robustness, flexibility or automation is desired.

On the other side, direct methods have the advantage that the user does not have to be concerned with deriving the first-order necessary conditions. Furthermore, direct methods are easier to initialize due to a larger domain of convergence and the physically intuitive meaning of the optimization variable. Although they still rely on a tentative guess and may not converge to the optimal solution, direct methods find at least a suboptimal solution unlike indirect approaches. This fact may be useful for concurrent engineering teams. Another point of success of direct methods is that even complex control or state constraints can be handled easily and that, in case of path inequality constraints, the sequence of free and constrained arcs does not need to be known a priori. As a major drawback, with a direct method is always uncertain whether the trajectory found by solving the NLP is truly an optimal solution to the original COCP or a suboptimal one.

Dynamic programming has two main advantages when compared to all other methods presented. First, the whole state space is searched; thus, an optimal solution is also the global optimum. Second, all controls are precomputed once a solution is found. This implies that closed-loop control policies instead of an open-loop control trajectory can be obtained, as well as it can be naturally extended to tackle uncertain and stochastic problems. The main drawback of dynamic programming relies on the curse of dimensionality. Therefore, memory and computational times of standard dynamic programming grow quickly with the number of state variables and become impractical for high-dimensional state space. The direct application of dynamic programming is therefore limited in practice to problems with low state-space dimensionality. Notably, the curse of dimensionality is resolved when using approximated techniques, based on local approximations of the value function, such as Differential Dynamic Programming. However, the obtained solution is no longer guaranteed to be globally optimal and the closed-loop control is only locally valid.

Regarding the solution approaches, gradient-based approaches provide deterministic conditions for convergence. They are able to handle a large number of problem variables and constraints. However, they require the constraint and objective function to be twice

differentiable. Consequently, they are not well suited for problems that use tabular data, or suffer from discontinuities. These methods require the user to provide an initial guess in the neighborhood of the initial guess. Gradient-Based solvers also find problems when searching for the global solution over wide design spaces and are not able to explore multiobjective design space in one run. Heuristic methods are well suited for problems with a reduced number of variables but with a high-dimensional space. They do not require an initial guess, which facilitates the automation of the process. While a gradient method is a local method a heuristic method is a global technique. These methods are more flexible, since they do not require the involved functions to be differentiable. However, when using heuristic algorithms, it is always uncertain if the obtained solution is optimal, since no optimality conditions are applied. In fact, in every run, a different solution can be obtained. Moreover, constraints are difficult to be met, since no gradient information is exploited. Hybrid approaches exhibit intermediate performances in terms of flexibility, robustness and optimality with respect to deterministic and heuristics methods. They exploit some features of the heuristic methods: being automatable, handling integers variable, and efficiently searching over large and multiobjective design space; and compensates their bad constraint handling capability by using a gradient-based method.

Qualitative comparison of dynamic programming, direct methods, and indirect approaches, along with gradient-based and heuristic solutions for solving continuous optimal control problems is shown in Figure 7 in terms of three criteria: flexibility, robustness and optimality. The green color means high performance on the selected criteria, the red color means poor performance, whereas orange implies intermediate performance. For example, direct methods exhibit high flexibility and robustness, whereas dynamic programming is more suitable when seeking for optimality and robustness. Regarding numerical solution approaches, hybrid methods provide a good compromise between optimality, robustness, and flexibility, when compared to purely heuristic or gradient-based solutions. Therefore, the best combination of numerical approaches and solutions for concurrent engineering may imply a direct approach with a hybrid solution technique.

| | | Flexibility | Robustness | Optimality |
|---|---|---|---|---|
| **Numerical Approaches** | **Indirect** | 🟥 | 🟥 | 🟩 |
| | **Direct** | 🟩 | 🟩 | 🟥 |
| | **Dynamic Programming** | 🟥 | 🟩 | 🟩 |
| **Numerical solutions** | **Deterministic** | 🟥 | 🟩 | 🟩 |
| | **Heuristic** | 🟩 | 🟥 | 🟥 |
| | **Hybrid** | 🟧 | 🟧 | 🟧 |

**Figure 7.** Methods and techniques in numerical approaches.

## 7. Approaches and Solutions for HOCPs

Numerical approaches to solve HOCPs are also categorized as dynamic programming, direct methods, or indirect methods. They inherit all of the positive and negative aspects from their application to COCP [11]. However, optimal control for hybrid systems is challenging due to the close interconnection of continuous and discrete dynamics. Methods for COCPs problems are not able to handle HOCPs since discrete decisions influence the continuous optimization. Similarly, methods for purely discrete optimization problems are unsuitable since the discrete optimization strongly depends on the continuous optimal control. Combining methods from COCP and discrete optimization is not straightforward. Continuous optimal control relies on infinitesimal variations of control and state variables and derivatives of functions. Such concepts are difficult to translate to discrete decision problems. In contrast, discrete optimization often relies on graph based search

methods, which are not applicable for continuous optimal control problems as these are infinite dimensional.

The first-order necessary optimality conditions for HOCPs are provided by the so-called hybrid minimum principle in (HMP) [38], which is generalization of the PMP for control systems with both continuous and discrete states and dynamics. It includes state and adjoint differential equations, a minimization of the Hamiltonian with respect to the continuous control, initial and terminal conditions for the state and/or adjoint variables, jump conditions for the adjoint variables, and Hamiltonian value conditions specifying the optimal discrete event times. However, no condition with respect to the sequence of discrete events can be given. This fact would imply that the sequence of gravity assists, of electric engine on/off switchings, or the optimal sequence of discrete states (e.g., launcher, thruster) have to be provided by the user. For this case, the HMP converts the HOCP into a MPBVP, which can be solved applying indirect shooting, collocation or gradient-methods [38]. Dynamic programming theory has been extended in [39] to tackle general classes of HOCPs, which in fact can be solved with DDP techniques. Though several algorithms have been developed, the convergence of the approximated value function to the true value function is in general still to be shown [38].

Therefore, HOCPs are typically solved with direct methods. They are usually formulated as Mixed-integer Nonlinear Programming (MINLP), i.e., NLPs where the optimization variables may be real or discrete. If the discrete state is identified with a finite sequence of phases and the discrete control can be described by an integer variable, then the HOCP can be converted to a MINLP by applying direct single/multiple shooting or collocation, where the continuous/discrete controls are discretized/parametrized. The solution to MINLPs has been shown to be NP-hard to solve [40], i.e., it is "at least as hard as any NP-problem". Therefore, various methods have been developed to reduce the computational time. The most prominent method in hybrid spacecraft trajectory optimization consists on a hybrid scheme with two-nested optimization loops. The inner loop solves for the continuous variable with a gradient-based solver, and the outer loop handles the discrete variables with a heuristic algorithm. Other methods include: branch and bound, branch and cut, outer approximation, or the generalized Benders decomposition [41].

## 8. Existing Low-Thrust Optimization Tools

The preliminaries required for formulating and solving low-thrust trajectory optimization problems have been briefly explained through previous sections. Hereafter, an overview of existing and representative low-thrust trajectory optimization tools and research works will be presented. The main goal is to summarize and review their main characteristics, capabilities and limitations, in order to identify which are the research gaps to advance into tools that could be used in concurrent engineering teams. First, analytical solution approaches will be presented and followed by indirect, direct and dynamic programing methods. A special section is dedicated to analyze the methods that implements predefined control laws applied within direct methods schemes. A total of 90 references have been investigated, among which 18 correspond to analytical solutions methodologies, while the remaining 72 are numerical approaches. Numerical approaches corresponding to indirect, direct, predefined control laws and dynamic programming have been summarized in Tables 1, 2, 3, and 4 respectively. They include information about the name of the tool, the developing company, organization or author, the type of numerical approach, objective, dynamics, state vector, and application.

**Table 1.** Representative Tools Implementing Indirect Methods for Low-Thrust Trajectory Optimization.

| Name | Ref | Company/Org./Author | Approach | Solution | Obj. | Dynamics | States | Transfers |
|------|-----|---------------------|----------|----------|------|----------|--------|-----------|
| VARITOP | [42] | JPL | Single Shooting | GB | SO | PR-TBP | CSV | IT |
| SEPTOP | [43] | JPL | Single Shooting | GB | SO | PR-TBP | CSV | IT |
| NEWSEP | [44] | JPL | Single Shooting | GB | SO | PR-TBP | CSV | IT |
| SAIL | [45] | JPL | Single Shooting | GB | SO | PR-TBP | CSV | IT |
| HILTOP | [46] | SpaceFlight Sol. | Single Shooting | GB | SO | PR-TBP | CSV | IT |
| ETOPH | [47] | CNES | Single Shooting | GB | SO | PR-TBP | CSV | IT |
| ITOP | [48] | Aerospace Corp. | Single Shooting | GB | SO | PR-TBP | MEE | PC |
| LT20 | [49] | Milano Univ. | Single Shooting | GB | SO | PR-TBP | MEE | PC |
| Tfmin | [50] | CNES | Single Shooting | GB | SO | PR-TBP | COE | PC |
| - | [51] | Kéchichian | Single Shooting | GB | SO | PR-TBP | MEE | PC |
| T-3D | [52] | Thales | Single Shooting | GB | SO | PR-TBP+AVG | MEE | G |
| SOFTT | [53] | Thales | Single Shooting | GB | SO | PR-TBP+AVG | - | PC |
| ELECTRO | [54] | OHB | Single Shooting | GB | SO | PR-TBP+AVG | MEE | PC |
| MIPELEC | [55] | CNES | Single Shooting | GB | SO | PR-TBP+AVG | MEE | PC |
| SEPSPOT | [56] | NASA | Single Shooting | GB | SO | PR-TBP+AVG | MEE | PC |
| GA-SEPTOP | [57] | JPL | Single Shooting | HY | MO | PR-TBP | CSV | IT |
| LOTTO | [58] | SES Engineering | Single Shooting | GB | SO | PR-TBP | MEE | PC |
| - | [59] | Torino Univ. | Single Shooting | HS | SO | PR-TBP | CSV | IT |
| - | [60] | Pontani et al. | Single Shooting | HS | SO | PR-TBP | PSV | IT |
| - | [61] | Lee et al. | Single Shooting | HS | MO | PR-TBP | CSV | IT |
| BNDSCO | [62] | Hamburg. Univ | Multiple Shooting | HS | SO | - | - | G |
| LOTNAV | [63] | Deimos Space | Multiple-shooting | GB | SO | CSV | PR-NBP | IT |
| - | [64] | Meng et al. | Multiple-Shooting | GB | SO | PR-TBP | MEE | PC |
| - | [65] | Olympio | Gradient method | - | SO | PR-NBP | PSV | G |

GB = Gradient-Based, HS = Heuristic, HY = Hybrid, SO = Single-Objective, MO = Multiobjective, IT = Interplanetary, PC = Planetocentric, G = General, SM = Stark-Model, KM = Kepler-Model, AVG = Averaging, AN = Analytical, CSV = Cartesian-State-Vector, MEE = Modified-Equinoctial-Elements, COE = Classical-Orbital-Elements, PSV3 = Cylindrical-Coordinates, PR = Perturbed-Restricted, TBP = Two-Body-Problem, NBP = N-Body Problem.

The yearly distribution for the publication dates of the examined references is shown in Figure 8. It can be seen that half of the references has been published in the last decade. Notably, among the analyzed numerical methods, direct methods represent a 65%, while indirect and dynamic programming are the 30% and 5%, respectively. The most widely implemented direct method is the single-shooting algorithm (38%), followed by collocation (32%), multiple-shooting (18%), and differential inclusion (2%). Similarly, the most common indirect method is single shooting (86%), followed by multiple-shooting (9%) and gradient methods (5%). Remarkably, a 75% of the numerical solution approaches use a gradient-based solver to tackle the resulting mathematical problem, while a 20% use purely heuristic algorithms and the remaining 5% apply hybrid algorithms. Finally, most approaches have been dedicated to solve single-objective problems (83%), while the remaining 17% exhibit the capability of solving multiobjective optimization problems. These statistics are illustrated in Figure 9.

**Table 2.** Representative Tools Implementing Direct Methods for Low-Thrust Trajectory Optimization.

| Name | Ref | Company/Org./Author | Approach | Solution | Obj. | Dynamics | States | Transfers |
|---|---|---|---|---|---|---|---|---|
| ASTOP | [66] | Space Flight Solutions | Single Shooting | GB | SO | PR-NBP | CSV | IT |
| COPERNICUS | [67] | Texas Univ., JSC | Multiple Shooting | GB | SO | PR-NBP | CSV | G |
| jTOP | [68] | Tokio Univ., JAXA | Multiple Shooting | GB | SO | PR-NBP | CSV | G |
| DITAN | [69] | ESA, Milano Univ. | Collocation | GB | SO | PR-NBP | CSV | G |
| MODHOC | [70] | Strathclyde Univ. | Collocation | HY | MO | PR-NBP | CSV | G |
| DIRETTO | [71] | Milano Univ. | Collocation | GB | SO | PR-NBP | CSV | G |
| MAVERICK | [72] | Colorado Boulder Univ. | Collocation | GB | SO | PR-NBP | CSV | G |
| MColl | [73] | NASA. | Collocation | GB | SO | PR-NBP | CSV | G |
| COLT | [74] | Purdue Univ. | Collocation | GB | SO | PR-NBP | CSV | G |
| GMAT | [75] | NASA | Collocation | GB | SO | - | - | G |
| STK | [76] | AGI | Collocation | GB | SO | - | - | G |
| OTIS | [77] | GCR, Boeing | Collocation | GB | SO | - | - | G |
| POST | [78] | NASA | Single Shooting | GB | SO | - | - | G |
| SOCS | [79] | Boeing | Collocation | GB | SO | - | - | G |
| DIDO | [80] | TOMLAB | Collocation | GB | SO | - | - | G |
| GPOPS | [81] | Univ. of Florida | Collocation | GB | SO | - | - | G |
| OPTELEC | [82] | Airbus | Multiple Shooting | GB | SO | PR-TBP | MEE | PC |
| MANTRA | [83] | ESA | Multiple-shooting | GB | SO | PR-NBP | CSV | G |
| LOTOS | [84] | ASTOS Solutions | Collocation | GB | SO | PR-TBP | MEE | PC |
| XIPSTOP | [85] | Boeing | Collocation | GB | SO | PR-TBP | MEE | PC |
| GALLOP | [86] | JPL,Purdue Univ. | Multiple-Shooting | GB | SO | KM | CSV | IT |
| COLTT | [87] | Colorado Boulder | Multiple-Shooting | GB | SO | KM | CSV | IT |
| LInX | [88] | J.H. Univ., Nabla Zero | Multiple-Shooting | GB | SO | KM | CSV | IT |
| BOLTT | [89] | Colorado Boulder | Multiple-Shooting | GB | SO | KM | CSV | IT |
| MALTO | [90] | JPL | Multiple-Shooting | GB | SO | KM | CSV | IT |
| EMTG | [91] | GSFC, Illinois Univ. | Multiple-Shooting | HY | MO | KM | CSV | IT |
| PaGMO | [92] | ESA | Multiple-Shooting | HY | SO | KM | CSV | IT |
| GA-GALLOP | [93] | Purdue Univ. | Multiple-Shooting | HY | MO | KM | CSV | IT |
| - | [94] | Zuiani et al. | Multiple-Shooting | GB | SO | SM | CSV | IT |
| DIFINC | [95] | Coverstone et al. | Differential Inclusion | GB | SO | PR-TBP | CSV | IT |
| - | [96] | Gerald et al. | Single Shooting | HS | SO | PR-TBP | PSV | IT |
| - | [97] | Pontani et al. | Single Shooting | HS | SO | PR-TBP | PSV | IT |

GB = Gradient-Based, HS = Heuristic, HY = Hybrid, SO = Single-Objective, MO = Multiobjective, IT = Interplanetary, PC = Planetocentric, G = General, SM = Stark-Model, KM = Kepler-Model, AVG = Averaging, AN = Analytical, CSV = Cartesian-State-Vector, MEE = Modified-Equinoctial-Elements, COE = Classical-Orbital-Elements, PSV3 = Cylindrical-Coordinates, PR = Perturbed-Restricted, TBP = Two-Body-Problem, NBP = N-Body Problem.

From the presented statistics, preliminary conclusion can be derived regarding the available tools that could be suitable for being used in concurrent engineering environments. First, only 17% of the tools are able to explore multiobjective design spaces, which prevents mission designers from obtaining a complete overview of the search space. Secondly, hybrid algorithms, which have been identified by the authors to be the most suitable for the preliminary design, has been incorporated in a 5% of the analyzed reference. In fact, in this section t will be shown that the goal of the developed hybrid tools are to increase automation, flexibility, and speed, at the cost of accuracy and optimality.

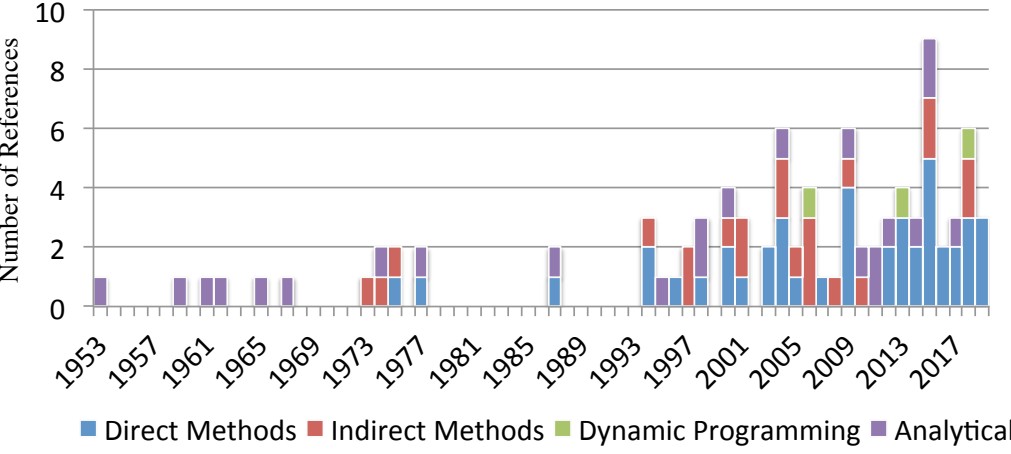

**Figure 8.** Illustration of the problem statement.

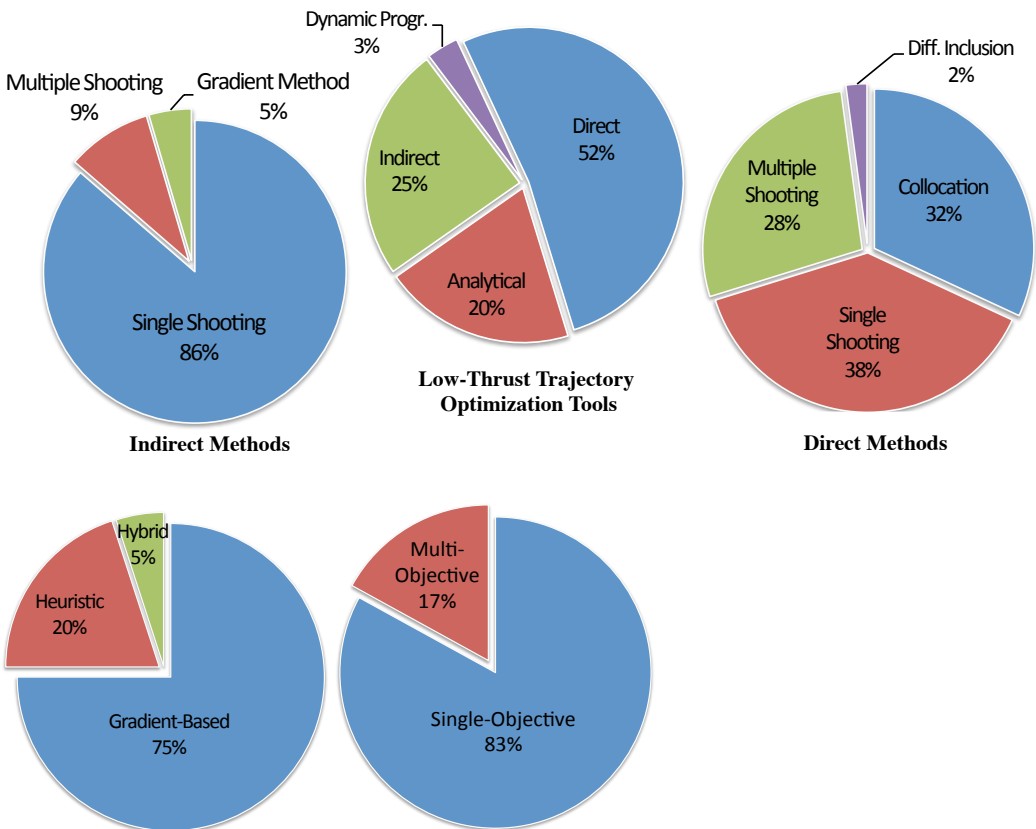

**Figure 9.** Overview of investigated Low-Thrust Optimization tools.

*8.1. Analytical Solutions*

There have been valuable efforts to solve simple low-thrust trajectory cases analytically. For instance, by either fixing the direction of the thrust, e.g., constant tangential or radial thrust, or by simplifying the boundary conditions, e.g., solving coplanar circle-to-circle transfers. They are convenient for rapidly evaluating low-thrust trajectories, or to be combined with a numerical optimization technique, either as an initial guess or as a dynamical model. One of the first pioneers in the history of analytical solutions was Tsien [98]. In his work of 1953, which has been exquisitely reproduced by Battin [99], analytical approximated planar solutions are derived in case of radial and circumferential thrust for initially circular orbits. An alternative closed-form solution in terms of an orbital anomaly and elliptic functions was derived by Izzo et al. [100]. Bombardelli et al. [101] and Gonzalo et al. [102] proposed a first-order asymptotic solution for the trajectory in the case of constant tangential and radial acceleration, respectively. Exact solutions to the tangential thrust problem have eluded researchers, but explicit solutions for certain variables can be found. Expressions defining the escape conditions or the amplitude of the bounded motion have been provided by Prussing et al. [103] and Mengali et al. [104].

In 1961 Edelbaum's [105] original analysis involved a low-thrust transfer between two circular orbits with a constant out-of-plane angle. He derived analytical expressions for the total velocity change and time of flight, and served as a starting point for many subsequent analysis. Later, Kéchichian [106] reformulated Edelbaum's problem [105] by applying optimal control theory to the minimum-time transfer problem to obtain the optimal time varying semimajor axis, inclination and yaw angles. Edelbaum [107] provided a complete first-order asymptotic solution for the Hamiltonian system resulting from power-limited transfer between coplanar elliptic orbits of arbitrary size and orientation. Fernandes et al. [108] obtained a first-order analytical solution, which includes short periodic terms, of the resulting average Hamiltonian system resulting from the optimal low-thrust transfers between coplanar orbits with small eccentricities. Zuiani et al. [94] presented a first-

order analytical solution for general transfers. He exploits the benefits of using a set of nonsingular orbital elements.

**Table 3.** Representative Tools Implementing Direct Methods with Predefined Control laws for Low-Thrust Trajectory Optimization.

| Name | Ref | Company/Org./Author | Approach | Solution | Obj. | Dynamics | States | Transfers |
|---|---|---|---|---|---|---|---|---|
| HYTOP | [109] | Aerospace Corp. | Blended Control | GB | SO | PR-TBP | MEE | PC |
| - | [110] | Yang Gao | Blended Control | GB | SO | PR-TBP+AN+AVG | COE | PC |
| - | [111] | Yang Gao | COV-Based | GB | SO | PR-TBP+AVG | MEE | PC |
| - | [112] | Strathclyde Univ | Blended Control | HY | MO | SM+AVG | COE | PC |
| SEPDOC | [113] | Kluever et al. | Blended Control | GB | SO | PR-TBP+AVG | COE | PC |
| - | [114] | Hudson et. al | Fourier-Expansion | GB | SO | PR-TBP+AN+AVG | COE | PC |
| - | [115] | Chang et. al | Lyapunov Control | GB | SO | PR-TBP | CSV | PC |
| LATOP | [116] | ESA | Lyapunov Control | HS | MO | PR-TBP | MEE | PC |
| GA-Q-Law | [117] | JPL | Lyapunov Control | HS | MO | PR-TBP | MEE | PC |
| STOUR-LTGA | [118] | JPL, Purdue Univ. | Shape-based | HS | SO | PR-TBP+AN | PSV | IT |
| IMAGO | [119] | Pascale et al. | Shape-based | HS | SO | PR-TBP+AN | MEE | IT |
| - | [120] | Wall et al. | Shape-based | HS | SO | PR-TBP+AN | PSV | IT |
| - | [121] | Taheri et al. | Shape-based | HS | SO | PR-TBP+AN | PSV3 | IT |
| - | [122] | Gondelach et al. | Shape-based | HS | SO | PR-TBP+AN | PSV3 | IT |
| - | [123] | Roa et al. | Shape-based | HS | SO | PR-TBP+AN | PSV | IT |
| MOLTO-IT | [22] | Morante et al. | Shape-based | HY | MO | PR-TBP+AN | PSV | IT |
| MOLTO-OR | [124] | Morante et al. | Lyapunov Control | HS | MO | PR-TBP | MEE | PC |
| InTrance-GA | [125] | DLR | Neural control | HY | SO | PR-TBP | CSV | IT |

GB = Gradient-Based, HS = Heuristic, HY = Hybrid, SO = Single-Objective, MO = Multiobjective, IT = Interplanetary, PC = Planetocentric, G = General, SM = Stark-Model, KM = Kepler-Model, AVG = Averaging, AN = Analytical, CSV = Cartesian-State-Vector, MEE = Modified-Equinoctial-Elements, COE = Classical-Orbital-Elements, PSV3 = Cylindrical-Coordinates, PR = Perturbed-Restricted, TBP = Two-Body-Problem, NBP = N-Body Problem

**Table 4.** Representative Tools Implementing Dynamic Programming for Low-Thrust Trajectory Optimization.

| Name | Ref | Company/Org./Author | Approach | Solution | Obj. | Dynamics | States | Transfers |
|---|---|---|---|---|---|---|---|---|
| MYSTIC | [126] | NASA | DDP | - | SO | PR-NBP | CSV | G |
| - | [127] | Colorado Boulder Univ. | DDP | - | SO | PR-TBP | MEE | PC |
| HDDP | [128] | Lantoine et al. | HDDP | - | SO | SM/KM | CSV | G |

GB = Gradient-Based, HS = Heuristic, HY = Hybrid, SO = Single-Objective, MO = Multiobjective, IT = Interplanetary, PC = Planetocentric, G = General, SM = Stark-Model, KM = Kepler-Model, AVG = Averaging, AN = Analytical, CSV = Cartesian-State-Vector, MEE = Modified-Equinoctial-Elements, COE = Classical-Orbital-Elements, PSV3 = Cylindrical-Coordinates, PR = Perturbed-Restricted, TBP = Two-Body-Problem, NBP = N-Body Problem.

Ruggiero et al. [129] developed analytical solutions for the optimal steering angles that maximize the instantaneous change of each COE independently. In [130], Kéchichian derived analytical solutions for transferring between circular orbits for two different scenarios: for the simultaneous change of semimajor axis and inclination, and for changing the argument of the ascending node and the semimajor axis. Burt [131] presented closed-form analytical formulas to compute the velocity increment and trip time for adjusting the eccentricity at a constant semimajor axis. This is accomplished with a constant in-plane acceleration perpendicular to the apsided line. Pollard [132] extended Burt's approach to the case of discontinuous acceleration by analyzing the perigee-and apogee-centered burn arcs, and extended the analysis to simultaneously change the eccentricity and inclination. Many of the aforementioned analytical approaches are implemented in the preliminary design software tool CAMELOT (Computational–Analytical Multifidelity Low-thrust Optimization Toolbox) [133].

There exists some trajectory analytical results for transfers incorporating Earth environmental effects. For instance, Kéchichian [134] obtained analytical solutions under the assumption of constant tangential thrust. He included the effect of $J_2$ and engine shut down during eclipses along small-to-moderate eccentricity orbits in terms of non-singular elements. Kluever [135] included periods of zero thrusting due to the Earth shadow eclipses and develop a semianalytical algorithm to solve the Edelbaum's problem. Kechichian [136] and Colasurdo et al. [137] also developed a purely analytical method for obtaining low-thrust and multirevolution transfers between coplanar circular orbits in the presence of Earth shadow, constraining the eccentricity to remain zero during the transfer. A two-variable asymptotic expansion method applicable to transfers from elliptic orbits was considered by Flandro [138], who included shadow penalty terms due to eclipses. Gao [110] obtained analytical solutions of the averaged equations when a predefined control law is applied, including shadow and $J_2$.

### 8.2. Indirect Methods

The most common indirect method is the indirect single shooting. It has been implemented in the tools SEPTOP (Solar Electric Propulsion Trajectory Optimization Program) [43], VARITOP (VARIational calculus Trajectory Optimization Program) [42], NEWSEP (NEW Solar Electric Propulsion trajectory optimization program) [44], and SAIL [45]. These tools have been developed at the Jet Propulsion Laboratory (JPL) and they are part of the Low-Thrust Trajectory Tool Suite (LTTT). The most general of the suite is VARITOP, which handles nuclear electric propulsion as well as solar electric propulsion and sail trajectories. However, solar electric engines and solar sails are more accurately modeled in the SEPTOP and SAIL programs, respectively. NEWSEP is a variation of SEPTOP that can accept discrete values of a thruster's throttle table rather than approximating the polynomial as its predecessor. They have been extensively used to design a variety of missions. For instance, NEWSEP provided trajectory support for the Deep Space 1 mission [45]. Runtimes for these tools range from hours to days [45], especially for those trajectories with numerous intermediate flybys.

Single shooting algorithms were also implemented in the tools HILTOP (Heliocentric Interplanetary Low Thrust Optimization Program) [46] and ETOPH (Electric Transfer Optimization with Planetocentric and Heliocentric phases) [47]. HILTOP was employed in numerous NASA and industry studies of missions to most planets, comets and asteroids. This tool lead to the development of MAnE-EP (Mission Analysis Environment for Electric Propulsion), which is an updated version of HILTOP. The tool ETOPH incorporates a smoothing technique for overcoming the difficulty of predefining the sequence of active constraints, and to reduce the numerical instabilities associated with the bang-bang structure of the control. The tool LOTNAV (Low-Thrust Interplanetary Navigation Tool), which implements an indirect single-shooting algorithm, has been the reference tool for ESA in the design of finite-thrust and ballistic interplanetary spacecraft trajectories and in the preliminary assessment of navigation and guidance issues on the computed trajectories. Aforementioned tools implement a patched two-body dynamics with CSV. Therefore, they are well suited for solving interplanetary trajectories, requiring the user to provide the flyby sequence, yet not for orbit-raising trajectories. They use a gradient-based solver and require an initial guess that is typically difficult to obtain.

Previous limitations are surmounted by using heuristic or hybrid techniques. Pontani and Conway [60] employed a PSO algorithm to solve an Earth-Mars rendezvous problem. They ignored the transversality conditions, as the objective function was optimized by the PSO and the constraints on the final state were included as penalties. A similar technique was presented by Lee et al. [61]. They combined a GA with simulated annealing to obtain trade-offs between delivered mass and required flight time for two-body and a three-body orbit transfers. Coverstone et al. [57] used a multiobjective GA to choose initial guesses for SEPTOP and optimized with respect to delivered mass, flight time and number of revolutions for an Earth-Mars rendezvous mission. Rosa and Casalino [59] employed a

GA to search for the combination of unknown parameters that minimizes the error on the boundary conditions; the minimum-error combination was provided as a guess to a gradient-based solver to obtain a converged solution. The procedure was tested in direct and multiple-gravity-assist missions to Mars.

Previous single shooting methods are not able to analyze planet-centered trajectories beyond a simple escape or capture maneuver, mainly because the EOM are expressed in CSV. Therefore, single shooting methods with MEE or COE have been developed. In [51], Kéchichian analytically derived the Hamiltonian system in terms of nonsingular elements without additional perturbations than a constant thrust acceleration. He solved for the unknown initial costates for a LEO-GEO transfer using a deterministic solver. The initial guess was obtained by setting to zero the values of the initial costates. A similar approach was implemented in the software tool Tfmin [50]. However, the technique from Kéchichian allows to rendezvous in the target orbit, while Tfmin was developed for free final longitude. Later, Kéchichian [139] extended his approach to account for the effect of $J_2$ perturbation, derived the set of dynamical and adjoint equations, and solves it for a LEO-GEO case. The initial guess was obtained by solving the problem without the oblateness effect. Kéchichian [140] further developed the low-thrust rendezvous in equinoctial elements by considering Earth zonal harmonics up to $J_4$.

However, previous approaches neither account for switching off the engine during eclipse, nor include coasting periods to obtain minimum-fuel consumption trajectories. For such purpose, software tools such as ITOP (Indirect Trajectory Optimization Program) [48], LT2O (Low-Thrust Trajectory Optimizer) [49], and LOTTO [58] were developed. They all are high-fidelity tools capable of solving min-time or min-fuel orbit transfers by implementing a switching function. They include eclipses, nonspherical Earth potential, solar radiation pressure, third-body perturbations, drag force, and altitude constraints via penalty functions. LOTTO further include slew rate restrictions and longitude targeting. Notably, ITOP was used for designing the electric orbit-raising maneuvers for the Al Yah 3 satellite [48]. ITOP and LT2O use gradient information to solve for the unknown initial costates. On the contrary, LOTTO uses a robust heuristic search method without relying on an initial guess. It selects the initial values for the costates that minimizes the error on the final constraints.

Accurately integrating the trajectory for the indirect shooting method is time-consuming due to the nonlinearities in the dynamics, the long flight-times and the high number of orbital revolutions. Thus, many authors have taken advantage of orbital-averaging techniques to greatly increase the speed of computation at the expense of fidelity. One of the most known softwares is SEPSPOT (It was previously named SECKSPOT (Solar Electric Control Knob Setting Program by Optimal Trajectories)) (Solar Electric Propulsion Steering Program for Optimal Trajectories) [56]. It was developed in the mid-1970's by Edelbaum et al. [141] to solve minimum-time transfers with a set of nonsingular elements. The program includes options for oblateness, shadowing with or without delay in thruster startup, an analytic radiation and power degradation model, and altitude constraints as penalties. However, the convergence probability is greatly diminished when solar cell degradation effects are included. The program has the option to solve hybrid transfers. For the high-thrust stage, one or two impulses of fixed magnitude are included, and the initial orbit is assumed to be circular.

Other examples include ELECTRO (ELECtric propulsion TRajectory Optimisation) [54], MIPELEC (Satellite Positioning with Electric Propulsion) [55], T3D [52] and SOFTT (Space Optimal Finite Thrust Transfer) [53]. MIPELEC is based on the theory developed by Geffroy and Epenoy [55] to solve min-time orbit-raising transfers with MEE, without shadow or oblateness effects. It is initialized by a user-provided guess or by an planar analytical approximation. ELECTRO implements EOM based on MEE to solve min-time transfers, including shadow and oblateness effects. An arbitrary user-provided guess is transformed into a feasible guess by an initial restoration phase. T3D solves min-time and min-fuel transfers including coast arcs by a smoothing mechanism, third-body perturbations, solar radiation pressure, oblateness, atmospheric drag and eclipse effects. A continuation

method is implemented to run from an arbitrary guess. In SOFTT [53], the authors apply the averaging to the Hamiltonian and use the averaging theorem. The main difference between MIPELEC, T3D, and ELECTRO is that the true longitude is the independent variable instead of time, and SOFTT uses a nondimensional representation of the state.

The remaining indirect methods, namely multiple-shooting, collocation and gradient-based, have been less popular, yet also have provided successful results. For instance, the general-software tool BNDSCO [62] implements indirect multiple-shooting. Oberle and Grimm [62] applied it intensively to study Earth-Mars low-thrust transfers. Meng et al. [64] implemented an indirect multiple-shooting algorithm where the transversality conditions were ignored, and the EOM were expressed in MEE. The unknown costates and the objective function were optimized by a gradient-based solver. He successfully solved a transfer from GTO to GEO. Olympio [21] developed an indirect gradient-based method using second-order derivative information. He was able to automatically find gravity assists naturally exploiting the multibody dynamics including space and capture phases. He also applied it to design an orbit raising transfer from LEO to MEO. Finally, although indirect collocation methods have been used in other fields, the author has not found any example of its application to low-thrust trajectory optimization.

*8.3. Direct Methods*

A variety of methods for computing multigravity assisted interplanetary and Earth-orbit transfers in accurate dynamical models implement direct methods combined with gradient-based solvers: POST (Program to Optimize Simulated Trajectories) [78] and ASTOP (Arbitrary Space Trajectory Optimization Program) [66] implements single shooting, Copernicus [67] and jTOP [68] use multiple shooting, while others such as DITAN (Direct Interplanetary Trajectory Analysis), MODHOC (Multiobjective Direct Hybrid Optimal Control) [70], OTIS (Optimal Trajectories by Implicit Simulation) [77], GMAT (General Mission Analysis Tool) [75], DIRETTO (DIREct collocation tool for Trajectory Optimization) [71], MAVERICK [72], Mcoll [73], COLT (Collocation with Optimization for Low-Thrust) [87], SOCS (Sparse Optimal Control Software) [79], GPOPS (Gauss Pseudospectral Optimization Software) [81], DIDO (Direct and Indirect Dynamic Optimization) [80], and STK/Astrogator (Systems Tool Kit) [76] implement collocation methods.

Some previous approaches correspond to software tools specifically designed for optimizing low-thrust trajectories (e.g., DIRETTO, MAVERICK, Mcoll, COLT), while others are general-purpose products for solving OCPs that have been used for solving low-thrust transfer problems (e.g., DITAN, MODHOC, OTIS, SOCS, GPOPS, DIDO). Notably, MOD-HOC is able to automatically search over a multiobjective design space and to handle discrete variables. Others are general space mission analysis tools that have specific modules for low-thrust trajectory optimization (e.g., GMAT, STK). Additionally, the computational load make them unsuitable for the preliminary design. Finally, the general purpose tool MANTRA uses a multiple-shooting technique [83]. It is the ESOC flight dynamics manoeuvre optimization software capable of computing multiple gravity assist trajectories including impulsive and low-thrust manoeuvres subject to given mission constraints. They all have proven to be effective for the design of low-thrust transfers. For instance, MANTRA and DITAN were used to design the multiple-flyby trajectory for Bepicolombo [142], while jTOP was used for the the trajectory for the microspacecraft PROCYON [68]. They implement multibody dynamics, but require the user to provide the sequence of flybys as well as an appropriate initial guess to converge.

Consequently, faster tools were developed at the cost of fidelity. One of the most widely-used algorithms for interplanetary transfers is the Sims-Flanagan Transcription (SFT) scheme. It implements a multiple-shooting scheme, the analytical Kepler model for the control, and instantaneous flybys. Most known tools include: GALLOP (Gravity Assisted Low-Thrust Local Optimization Program) [86], COLTT (CCAR Optimal low-Thrust Tool) [87], LInX (Low-thrust Interplanetary eXplorer) [88], MALTO (Mission Analysis Low-Thrust Optimizer) [90], EMTG (Evolutionary Mission Trajectory Generator) [143], BOLTT

(Boulder Optimal Low-Thrust Tool) [89] and PaGMO (Parallel Global Multiobjective Optimizer) [92]. Solutions from these tools are usually used as initial guesses for higher-fidelity tools. For instance, MALTO and GALLOP provide initial guesses for Copernicus and OTIS, while EMTG's solutions were used to feed GMAT [144]. A similar approach to the SFT was developed by Zuiani et al. [94], yet implementing the analytical Stark model between the multiple-shooting nodes.

Some of the previous methods used hybrid solutions approaches to avoid the need for the user to provide a suitable initial guess. For instance, Vavrina and Howell [93] presented GA-GALLOP, a program that use a GA to automatically provide initial guesses for GALLOP and to explore the multiobjective design space in terms of flight time and final mass. It was applied to Mars and Jupiter missions including one flyby. Yam et al. [92] used monotonic basin hopping (MBH) to automatically feed PaGMO. The approach was applied to maximize the final mass on a mission to Mercury involving up to six flybys. However, the tool require the user to provide the flyby sequence. An automated solution for the number and sequence of gravity assists has been addressed by Englander and Conway [143] in EMTG. In their approach they combine two nested optimization algorithms. The outer loop uses a GA to select the flyby number and sequence while the inner loop solves the corresponding sequence of interplanetary legs using MBH along with the SFT scheme. The method was proven to automatically determine the flyby sequences that maximize the delivered mass for missions to Mercury, the asteroid belt, and Pluto. This methodology was also tested on multiobjective problems [91].

A different approach has been considered by Gerald and Converstone-Carrol [96], and by Pontani et al. [97], who only relied on population-based heuristic methods to find a solution of the direct shooting transcription resulting from planar low-thrust interplanetary transfers without flybys. The former implemented a GA to solve for the time-discretized thrust directional angles that minimize the transfer time for an Earth-Mars transfers, and that minimizes the fuel consumption for an Earth-Mercury trajectory. They included a binary optimization variable to determine wether the engine is in thrusting or coasting mode. Constraints on the final state have been applied as penalties in the objective function. The latter modeled the thrust steering law as a linear combination of B-Spline functions and used a particle swarm algorithm to optimize the parameters defining them. They claimed that despite its simplicity and intuitiveness, the particle swarm methodology proved to be quite effective in finding the optimal solution to orbital rendezvous optimization problems with considerable numerical accuracy.

Other available software tools are especially dedicated to solve minimum-time and minimum-fuel electric orbit-raising problem including operational constraints, such as LOTOS (Low-thrust Orbit Transfer Optimization Software) [84], XIPSTOP (Xenon Ion Propulsion System Trajectory Optimization Program) [85], and OPTELEC [82]. The tools LOTOS and XIPSTOP implement a direct collocation scheme combined with a gradient-based solver, while OPTELEC uses multiple-shooting with a gradient-based solver. All of them include the possibility of imposing eclipse or radiation constraints, slew rate and power consumption restrictions, slot targeting, avoidance of the GEO ring, Sun-angle or sensor pointing constraints. They implement a perturbed two-body dynamics along with accurate models for Earth Oblateness. They have proven to successfully solve numerous transfers to GEO. For example, XIPSTOP and OPTELEC are used to calculate the maneuvers for Boeing's and Airbus all-electric platforms, respectively. Notably, LOTOS and OPTELEC are able to compute hybrid transfers, where the chemical orbit-raising is followed by an electric phase.

The remaining class of direct approaches refers to differential inclusion. Only one algorithm was found by the author. The tool DIFINC (DIFferential INClusion) was presented in [95] by Coverstone and William to compute low-thrust trajectories in the two-body problem with cartesian coordinates. This formulation removes explicit control dependence from the problem statement thereby reducing the dimension of the parameter space of the resulting nonlinear programming problem. They presented three interplanetary trajectory

examples: an Earth-Mars constant specific impulse transfer, an Earth-Jupiter constant specific impulse transfer, and an Earth-Venus-Mars variable specific impulse gravity assist. The work was later extended by Hargens and Coverstone [145]. They implemented DIFINC in terms of the modified equinoctial orbital elements and applied it to solve several missions including both nuclear electric and solar electric propulsion systems. Results obtained showed good agreement with industry-standard software, such as VARITOP.

### 8.4. Predefined Control Laws

In this section, direct methods that have approximated the control law by predefined guidance schemes will be detailed. They yield to sub-optimal solutions but are faster. The first class of the investigated control laws implement the COV-based guidance, and includes HYTOP [109] (HYbrid Trajectory Optimization Program) and the work done by Gao [111]. The former was developed in 1994 by Ilgen, uses orbital averaging and can calculate time-optimal and minimum propellant orbit raising transfers, constrained by Earth shadowing and oblateness. The software has been also applied to obtain a wide range of maximum-payload transfers to GEO using combined-chemical-electric propulsion. It has been also used to provide initial guesses to the indirect optimization software ITOP. In the work presented by Gao [111], a multiple-shooting scheme combined with orbital averaging was used to solve a series of minimum-time LEO-GEO and GTO-GEO transfers were solved using MEE, oblateness and a cylindrical shadow model. Results showed good agreement with the unaveraged dynamics.

The second class of methods include BC. In 1998, Kluever and Oleson proposed SEPDOC [113] (Solar Electric Propulsion Direct Optimal Control), which includes three extremal laws for changing semimajor axis, eccentricity, and inclination. It includes averaging, power degradation, oblateness and shadow. It exhibits better convergence than SEPSPOT in typical minimum-time LEO-GEO and GTO-GEO transfers. A COE correction scheme was developed by Ruggiero et al. [129], including coasting arcs but neglecting environmental perturbations. Gao's [110] employed three types of steering laws: perigee-centered tangential, apogee-centered inertial, and piecewise constant yaw. He derived analytic expressions for the averaged EOM in COE including shadow, coasting, and $J_2$. The weighting parameters were optimized using a deterministic algorithm for min-time and min-fuel transfers. In [112] Zuiani et al. proposed two-tangential control laws for planar transfers: perigee and apogee centered. Parameters where optimized with a multiobjective GA with respect to the time of flight, engine operation time, time within the radiation belt, and longest eclipse duration.

Hudson and Sheeres [114] represent each component of the thrust acceleration as a Fourier series (FS) in eccentric anomaly, and then average EOM in COE over one orbit to define a set of secular equations. The equations are a function of only 14 of the thrust FS coefficients, regardless of the order of the original Fourier series. Thus the continuous control is reduced to a set of 14 parameters. She solved a targeting problem using a least square method to solve for the unknown coefficients. Then, Ko and Sheeres [146] identified minimal sets of six FS parameters to represent the perturbing acceleration effectively, instead of 14. Given the initial and desired final orbital state, a set of six FS coefficients can be computed analytically, and the required control accelerations can be constructed to achieve any orbital maneuver. The method was demonstrated in [147] on two types of low-thrust spiral maneuvers: a repositioning maneuver in GEO and a maneuver to simultaneously change orbit radius and inclination. Results were successfully used as an initial guess for the STK optimization engine.

A different approach utilizes closed form feedback control laws derived from Lyapunov functions. For instance, Ilgen [148] developed a Lyapunov guidance law based on MEE. Gao [111] used it as an initial guess for his COV-based method. Petropoulos [117] presented the Proximity Quotient guidance law (Q-Law), which is expressed in terms of MEE, implements shadow and oblateness effects, and a coasting mechanism without averaging. A multiobjective GA was used to optimize the free parameters and was im-

plemented in the tool GA-Qlaw. It proved to permit a rapid trade-off evaluation and to provide reasonable performance estimates for the preliminary design of planetocentric transfers [116,149]. Additionally, it was integrated into the high-fidelity tool Mystic [150] to assist in generating starting guesses. A formulation of the proximity quotient based on MEE was implemented in LATOP (Lyapunov control Aided Transfer Optimizer Program) and combined with a genetic algorithm. This Q-law was also used by Morante et al. [124] in the tool MOLTO-OR, to incorporate the optimization of the propulsion system along the trajectory optimization. Morante et al. used it as an initial guess for a collocation method where he applied various operational constraints (e.g., slot-synchronization, avoidance of the geostationary ring). Another well-known Lyapunov function was introduced and rigorously proved by Chang et al. [151]. The controller is expressed in CSV and was used by Betts [79] to generate initial guesses for a direct collocation scheme implemented in SOCS for transfers to GEO and Molniya orbits. Gurfil [152] developed a Lyapunov controller in terms of COE and used it to determine orbital transfer between elliptical orbits.

Some of the analyses may be described as shape-based, that is, the trajectory shape is directly assumed, with the requisite thrust computed a posteriori. Notably, the first shape-based method was the logarithmic spiral presented as early as 1950 by Forbes [153] and 1959 by Tsu [154] and Bacon [155]. A remarkable variant on the logarithmic spiral was given by Pinkham [156] and Lawden [157]. Pinkham's spiral can be used, for example, to escape from an initially circular orbit, or from any point on an elliptic orbit. Although Lawden's spiral was developed with transfer between two arbitrary states in mind, the spiral does not offer enough degrees of freedom to accomplish this. Therefore, despite the various analytic results available for the logarithmic spiral, the solution essentially has a constant flight path angle. In an attempt to correct these shortcomings, the exponential sinusoid was developed Petropoulos and Longuski [118], which has two parameters, apart from the scaling and phase parameters. Izzo [158] explored the potential of exponential sinusoids for solving the accelerated multirevolution Lambert's problem. These early works are extensively reviewed by Petropoulos and Sims [7].

In Ref. [118], Petropoulos and Longuski apply a broad search algorithm with pruning criteria along with exponential sinusoids to generate candidate trajectories for GALLOP. The technique was implemented in the software STOUR-LTGA (Satellite Tour Design Program for Low-Thrust Gravity-Assist trajectories), which automatically searches for low-thrust, gravity-assis trajectories using a heuristic broad search algorithm. The user has to specify a sequence of gravity assist bodies, a range of launch dates, and a range of launch velocities for trajectories, subject to various constraints, such as time of flight and propellant consumption limits. They solved a rendezvous mission to Ceres via a Mars flyby, and a flyby mission to Jupiter via Venus-Earth-Mars flybys. However, the cost estimated by exponential sinusoid methods does not properly estimate the optimal value. It is due to the fact that neither coasting nor rendezvous phases have been included in the model. Vasile et al. [159] study the optimality of the exponential sinusoid and concludes that this model is far from satisfying the necessary condition of optimality.

Later works include Wall and Conway [120], who modeled the trajectory as an inverse polynomial with unbounded tangential thrust. The advantage of this approach compared to Petropoulos and Longuski's is the possibility to satisfy all boundary conditions. A GA was used in both works to select the unknown launch date, the time of flight, and the number of heliocentric revolutions to optimize a multirendezvous asteroid problem. Wall [160] extended their approach to three dimensional case by using cylindrical coordinates. De Pascale and Vasile [119], Novak and Vasile [161], Taheri and Abdelkhalik [121], and Gondelach and Noomen [122] created ingenious three-dimensional shape-based models incorporating pseudo-equinoctial elements, spherical coordinates, finite Fourier series, and hodographic shaping, respectively. These approaches can handle boundary, time of flight and thrust constraints and were used to solve various rendezvous problems without intermediate flybys via grid search over the free parameters. In fact, the pseudo-equinoctial

approach was implemented in the tool IMAGO [119] (Interplanetary Mission Analysis Global Optimization), an successfully used as initial guess for DITAN.

Previous methods, except for the hodographic method, assumed tangential thrust. To improve the versatility of the solution, Roa et al. [162] found an entire new family of Generalized Logarithmic Spirals based on the thrust profile of the logarithmic spirals. Therefore, it is a planar shape with unbounded thrust levels. The flexibility of this approach was later improved by adding an additional degree of freedom in the solution [163] and modeling the transversal motion with a polynomial shaping approach [164]. By using a thrust-coast-thrust sequence for rendezvous legs, and thrust-coast sequence for flybys legs, he was able to solve a rendezvous problem to Ceres via Mars flyby. Recently, Roa et al. opted in [123] to use his shaped-based method together with a branch and prune algorithm for the direct exploration of the search space to generate as many candidate trajectories as possible for a multiple-flyby mission to Jupiter. However, in his approach he predefined the sequence of flybys and did not include coast arcs. Candidate trajectories were used as initial guesses for GALLOP. Later, Morante et al. [22] combine this shape-shaped method with a GA to automatically obtain the number and sequence of flybys for various interplanetary missions in the tool MOLTO-IT. He proved that the shaped based method was suitable for providing initial guesses for more accurate optimization algorithms.

The last class of predefined control laws explores artificial neurocontrollers. The tool InTrance (INtelligent spacecraft TRAjectory optimization using NeuroController Evolution) was designed by Dachwald [165] only for heliocentric single-phase trajectory optimization problems. InTrance was later extended by Carnelli et al. [125] to include intermediate gravity assisted maneuvers in InTrance-GA. Dynamics is expressed in terms of patched two-body problems, where the flybys are unpowered but not instantaneous. It implements an artificial neural network to act as neucontroller and combine it with evolutionary algorithms (a GA) to train the NC and to determine the optimal spacecraft steering strategy that minimizes the total transfer time. The targeting constraints are handled by penalizing the objective function. This combination is known as evolutionary neurocontrol. Results are presented for a Mercury rendezvous with a Venus gravity assist and for a Pluto flyby with a Jupiter gravity assist. Computing times were 11 h for the former case and 6 hours for the latter scenario. They found a good agreement with other software standards as IMAGO, GALLOP and DITAN.

*8.5. Dynamic Programming Methods*

Whiffen [126] presented the Static/Dynamic Control (SDC) algorithm, a class of Differential Dynamic Programming (DDP) method. It was implemented in the generic tools for high-fidelity trajectories Mystic. It implements multibody dynamics and is able to naturally obtain the optimal sequence of flybys, including escape, capture phases. The tool itself can be seen as the state-of the art for the design of low-thrust trajectories and it has been successfully used to design NASA's cancelled Jupiter Icy Moon Orbiter (JIMO) and also to design and navigate the NASA's DAWN discovery mission to asteroid Vesta and Ceres. Results from this algorithm has been published in numerous papers, such as [166,167]. However, Mystic uses a pure penalty method to account for the constrained violation, which may lead to unfeasible trajectories, slow convergence, or no convergence at all. Additionally, its application to solve multirevolution planetocentric transfers is limited by its computation time to about 250 revolutions [126]. Last but not least, it requires a good initial guess to run.

A faster yet less accurate algorithm was presented by Lantoine and Russell [128] and implemented in the tool HDDP (Hybrid Differential Dynamic Programming). It is an extension of the classic DDP algorithm that combines DDP with well-proven nonlinear mathematical programming. It exploits second-order derivative information, and includes two options for the Dynamical modeling: the Stark model and the Kepler model. In [168], Lantoine and Russell presented a maximum final mass Earth-Mars rendezvous transfer and a 17 revolution minimum-fuel Earth-orbit transfer. Computational times were 60 s

for the former, and 20 min for the latter. A more appropriate method for handling high revolutions was developed by Aziz [127]. He discretizes the trajectory in terms of MEE and the control schedule with respect to an orbit anomaly and perform the optimization with DDP. He included spherical gravity and third- body perturbations. He solved geocentric transfers up to 2000 revolutions. He was able to generate a Pareto front trading time-off flight and propellant mass, by independent runs of his single-objective algorithm within a matter of hours.

### 9. Conclusions

Most common existing low thrust trajectory optimizers are generally complex and difficult to incorporate into the simpler spacecraft system models used for concurrent engineering. Moreover, most of them are not able to include mission-planning or discrete optimization as part of the solution, since they typically rely on gradient-based methods. There is a lack of low-thrust trajectory optimization tools that can search over multiobjective design spaces. Hereby, a list of identified research gaps that have been identified as relevant subjects for future are summarized:

- Optimize alternative objectives: it has been seen that typically, either propellant mass or time-of-flight are optimized. However, mission designer may be interested into minimizing the radiation absorbed during the passage through the Van-Allen radiation belts to reduce the damage into the solar panel, or into minimizing the time-spent in eclipse. Additionally, when including spacecraft design along with the trajectory optimization, other performance indexes, such as spacecraft total mass or target on-station mass may have to be included.
- Reduce computational time: among the presented tools, GA-EMTG is able to automatically find the sequence of gravity assists for an interplanetary mission with respect to multiple-objectives, requiring minimal user-interaction, and providing medium fidelity solutions. However, computational times range from several hours to days . Therefore, faster assessments at the cost of fidelity and optimality are desirable.
- Extend the capability of preliminary design tools to include mission constraints: low-thrust trajectory optimization tools used for the preliminary design due to their speed, such as implementing predefined control laws, do not have the ability to impose important mission constraints, which may imply that the obtained trajectory is not feasible. Thus, advancing into the incorporation of constraints into such tools, either by a penalty function or by a different predefined control law, will significantly enhance the success during the preliminary design.
- Increase the efficiency of searching over wider design spaces: presented hybrid and heuristic tools are able to work for a limited combinatorial complexity of the problem. However, they are not well-suited for solving problems such as asteroid tours, debris-removal missions, or asteroid mining mission, where the are thousands of available options. Improving the capability of searching over this broad spaces will enable the of more ambitious low-thrust missions. A potential approach would be to develop dedicated heuristic algorithms able to efficiently optimize over large sequences os visited bodies (e.g., asteroids, debris), possibly incorporating artificial intelligence into the approach.

**Author Contributions:** Conceptualization, D.M., M.S., and M.S.R.; investigation, D.M..; writing—original draft preparation, D.M.; review and editing, M.S.R., and M.S.; supervision, M.S.R., and M.S. All authors have read and agreed to the published version of the manuscript.

**Funding:** This research received no external funding.

**Institutional Review Board Statement:** Not applicable.

**Informed Consent Statement:** Not applicable.

**Data Availability Statement:** Not applicable.

**Conflicts of Interest:** The authors declare no conflict of interest.

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
