# Peer review of "A Survey on Low-Thrust Trajectory Optimization Approaches"

_aerospace, doi:10.3390/aerospace8030088_

Round 1

Reviewer 1 Report

The paper is very long (46 pages) and very difficult to follow.

Even commendable, it is not clear what is the use of it. A scientific literature survey should be a synthesis of the state-of-the art on a narrow problem of interest, with high degree of novelty. The problem approached in the paper looks to have a very high degree of generality, rather than being a narrow and “hot” topic of research.

The approach tackled in the paper is more suited to a book than a scientific paper. The literature survey includes books and papers from 1939, 1962, 1965, 1967, 1968, 1972, 1973, 1979...

It is clear that the depth of the survey exceeds the limits of the “state-of the-art” and tries to present an exhaustive analysis of a very complex, yet general problem.

Thus, my concerns are not about the quality of the work, but if its place is in research journal. In my opinion this work should published as a book, not as a journal paper.

Reviewer 2 Report

The paper is well written,  but needs few corrections to improve the quality of the paper :

  1. Line-50, typo " As a rule, it can be stated....."
  2. Line-130, typo " interplanetary or general...."
  3. Line-165, Equation number not correct.
  4. Line-208, typo in the word "continuos"
  5. Line-258, word "this" need to be replaced with "these"
  6. Line-423, fix the sentence.
  7. Line-438, fix the word "and"
  8. Line-725, table number missing.
  9. Line-788, Tables 4-1 not consistent with the order of numerical approaches as mentioned in text.
  10. Line-793, Figure-8(a) is not referenced correctly.
  11. Lines-837, 838, 840, 841 - Not consistent with the syntax "semi-major axis"
  12. Line-1242, fix the word " interplanetary or general"

Reviewer 3 Report

I appreciate the Authors' contribution to the literature review on this topic, but this article looks more like a chapter in a monograph. A few conclusions:

  1. English is at times incomprehensible, in several places the translation is incorrect;
  2. I do not really understand what the purpose of this article is, the Authors also do not define it;
  3. It would be easier if the authors presented the structure of the article in the introduction - it would be clear what to expect in the study;
  4. The Authors write that "A special emphasis will be placed on the extension of the classical techniques to solve hybrid optimal control problems" - is it their contribution to science, or the evaluation of existing studies?
  5. I do not fully understand fig. 7. Moreover, the chapter should not be finished with a drawing or a table.
  6. The Authors present statistics, for example in Figure 9. What are the conclusions of these statistics, what does this mean for science, apart from knowing which methods were used more or less frequently? 

Round 2

Reviewer 1 Report

The paper is still very long and does not look like a research paper, but a book chapter.

I do not deny the scientific quality of the content, as a scientific analysis, I just wonder what it will be used to publish it as a journal paper, not as a chapter in a book.

The paper, although valuable in terms of content, does not meet in my opinion, especially due to its length, the conditions of a scientific paper to be published in a journal.

However, if the journal's policy is to accept synthesis papers of such length, then, in terms of quality, the paper may be accepted.

Reviewer 3 Report

I accept this paper in present form.